# Surfactant protein C peptides with salt-bridges ("ion-locks") promote high surfactant activities by mimicking the α-helix and membrane topography of the native protein

Frans J. Walther[1,2], Alan J. Waring[1,2,3,4], José M. Hernández-Juviel[1], Piotr Ruchala[3], Zhengdong Wang[5], Robert H. Notter[5] and Larry M. Gordon[1]

[1] Los Angeles Biomedical Research Institute at Harbor-UCLA Medical Center, Torrance, CA, United States of America
[2] Department of Pediatrics, David Geffen School of Medicine, University of California at Los Angeles, Los Angeles, CA, United States of America
[3] Department of Medicine, David Geffen School of Medicine, University of California at Los Angeles, Los Angeles, CA, United States of America
[4] Department of Physiology & Biophysics, School of Medicine, University of California, Irvine, CA, United States of America
[5] Department of Pediatrics, University of Rochester, Rochester, NY, United States of America

Corresponding author
Frans J. Walther, fjwalther@ucla.edu

## ABSTRACT

**Background.** Surfactant protein C (SP-C; 35 residues) in lungs has a cationic N-terminal domain with two cysteines covalently linked to palmitoyls and a C-terminal region enriched in Val, Leu and Ile. Native SP-C shows high surface activity, due to SP-C inserting in the bilayer with its cationic N-terminus binding to the polar headgroup and its hydrophobic C-terminus embedded as a tilted, transmembrane α-helix. The palmitoylcysteines in SP-C act as 'helical adjuvants' to maintain activity by overriding the β-sheet propensities of the native sequences.

**Objective.** We studied SP-C peptides lacking palmitoyls, but containing glutamate and lysine at 4-residue intervals, to assess whether SP-C peptides with salt-bridges ("ion-locks") promote surface activity by mimicking the α-helix and membrane topography of native SP-C.

**Methods.** SP-C mimics were synthesized that reproduce native sequences, but without palmitoyls (i.e., SP-Css or SP-Cff, with serines or phenylalanines replacing the two cysteines). Ion-lock SP-C molecules were prepared by incorporating single or double $Glu^--Lys^+$ into the parent SP-C's. The secondary structures of SP-C mimics were studied with Fourier transform infrared (FTIR) spectroscopy and PASTA, an algorithm that predicts β-sheet propensities based on the energies of the various β-sheet pairings. The membrane topography of SP-C mimics was investigated with orientated and hydrogen/deuterium (H/D) exchange FTIR, and also Membrane Protein Explorer (MPEx) hydropathy analysis. *In vitro* surface activity was determined using adsorption surface pressure isotherms and captive bubble surfactometry, and *in vivo* surface activity from lung function measures in a rabbit model of surfactant deficiency.

**Results.** PASTA calculations predicted that the SP-Css and SP-Cff peptides should each form parallel $\beta$-sheet aggregates, with FTIR spectroscopy confirming high parallel $\beta$-sheet with 'amyloid-like' properties. The enhanced $\beta$-sheet properties for SP-Css and SP-Cff are likely responsible for their low surfactant activities in the *in vitro* and *in vivo* assays. Although standard $^{12}$C-FTIR study showed that the $\alpha$-helicity of these SP-C sequences in lipids was uniformly increased with Glu$^-$–Lys$^+$ insertions, elevated surfactant activity was only selectively observed. Additional results from oriented and H/D exchange FTIR experiments indicated that the high surfactant activities depend on the SP-C ion-locks recapitulating both the $\alpha$-helicity and the membrane topography of native SP-C. SP-Css ion-lock 1, an SP-Css with a salt-bridge for a Glu$^-$–Lys$^+$ ion-pair predicted from MPEx hydropathy calculations, demonstrated enhanced surfactant activity and a transmembrane helix simulating those of native SP-C.

**Conclusion.** Highly active SP-C mimics were developed that replace the palmitoyls of SP-C with intrapeptide salt-bridges and represent a new class of synthetic surfactants with therapeutic interest.

## INTRODUCTION

Lung surfactant is a mixture of lipids and proteins that is critical for normal breathing due to its ability to prevent alveolar collapse during expiration by reducing alveolar surface tension to extremely low values. Surfactant is synthesized and secreted into the alveolar fluid by alveolar type II cells and consists of approximately 80% phospholipids, 10% neutral lipids, and 10% proteins (*Goerke, 1998*). Although dipalmitoyl phosphatidylcholine and phosphatidylglycerol constitute the principal phospholipid components in lung surfactant, its biophysical activities largely depend on the presence of the hydrophobic surfactant protein B (SP-B) and, to a lesser degree, the extremely hydrophobic surfactant protein C (SP-C) (*Walther et al., 2007a*). Pioneering surfactant therapy using bovine or porcine lung extracts, which consists of polar lipids and native SP-B and SP-C, has greatly improved the outcome of premature infants with neonatal respiratory distress syndrome (NRDS) (*Polin & Carlo, 2014*). On the other hand, replacement surfactant therapies using animal preparations have not been successful in clinical trials of pediatric and adult patients with acute lung injury (ALI) and acute respiratory distress (ARDS) (*Brower & Fessler, 2011*; *Willson et al., 2013*). A major goal of current research is to replace animal-derived formulations for the various surfactant deficiencies with total synthetics consisting of SP-B and SP-C mimics, produced by recombinant technology or peptide synthesis, and selected synthetic lipids (*Curstedt, Calkovska & Johansson, 2013*; *Walther et al., 2005*; *Walther et al., 2007b*; *Walther et al., 2010*; *Walther et al., 2014*; *Spragg et al., 2011*; *Almlen et al., 2010*).

**Amino-acid sequences of Human, Dog and Pig SP-C proteins and SP-C mimics**

| SP-C Protein or Mimic | Amino-acid sequence | | | |
|---|---|---|---|---|
| *SP-C proteins* | | | | |
| Numbering | 1 | 10 | 20 | 30 |
| Human SP-C | FGIPCCPVHL | KRLLIVVVV | VLIVVVIVGA | LLMGL |
| Dog SP-C | GIPCFPSSL | KRLLIIVVVI | VLVVVVIVGA | LLMGL |
| Pig SP-C | LRIPCCPVNL | KRLLVVVVV | VLVVVVIVGA | LLMGL |
| *SP-C mimics* | | | | |
| | 1 | 10 | 20 | 30 |
| rSP-C (Venticute®) | FGIPFFPVHL | KRLLIVVVV | VLIVVVIVGA | LLIGL |
| SP-Cff | GIPFFPVHL | KRLLIVVVV | VLIVVVIVGA | LLMGL |
| SP-Cff ion-lock 1 | GIPFFPVHL | KRLLIVVVV | **E**LIV**K**IVGA | LLMGL |
| SP-Css | GIPSSPVHL | KRLLIVVVV | VLIVVVIVGA | LLMGL |
| SP-Css ion-lock 1 | GIPSSPVHL | KRLLIVVVV | **E**LIV**K**VIVGA | LLMGL |
| | 1 | 10 | 20 | 30 |
| SP-C33 UCLA | IPSSPVHLK | RLKLLLLLL | LILLLILGAL | LMGL |
| SP-C33 ion-lock 1 | IPSSPVHLK | RLKLLLLLL | **E**ILL**K**ILGAL | LMGL |
| SP-C33 ion-lock 2 | IPSSPVHLK | RLKLL**K**LLL**E** | **E**ILL**K**ILGAL | LMGL |

**Figure 1 Amino-acid sequences (1-letter codes) for native SP-C proteins and SP-C mimic peptides.** Human, dog and pig SP-C are the native sequences (35 or 34 residues), with palmitoyl groups (*not shown*) attached to the Cys residues in the N-terminal domain. SP-C mimics are based on the native sequences, except that the palmitoylcysteines have been replaced by Phe or Ser residues. The SP-C mimic sequences have been slightly shifted to permit maximal overlaps with the SP-C protein sequences. rSP-C is a 35-residue, recombinant form of human SP-C, with Phe residues replacing Cys-4 and Cys-5 and Ile replacing Met-32; SP-C33 UCLA has the same sequence as the conventional SP-C33; SP-Cff ion-lock 1 is SP-Cff(E20/K24), SP-Css ion-lock 1 is SP-Css(E20/K24) and SP-C33 ion-lock 1 is SP-C33(E20/K24), each with an ion-lock between Glu⁻-20 and Lys⁺-24. SP-C33 ion-lock 2 is SP-C33(K15/E19; E20/K24), with ion-locks between Lys⁺-15 and Glu⁻-19 and between Glu⁻-20 and Lys⁺-24. Paired glutamates and lysines are underlined red residues, with salt-bridges represented as red lines. The SP-Cff and SP-Css mimics are each 34-residues long (except for rSP-C) and based on the native SP-C class, while those based on the SP-C33 class are 33-residues.

SP-C is a short (35 amino acids; MW of 4.2 kDa in humans) protein that is highly enriched in valine, leucine and isoleucine residues, making it much smaller and more hydrophobic than SP-B (79 amino acids; MW of 8.7 kDa) (*Walther et al., 2007a*). Mature SP-C has an N-terminus with a pair of vicinal cysteine residues that are covalently linked to palmitoyl moieties via thioester bonds (Fig. 1), thereby producing a true *proteolipid* (*Curstedt et al., 1990*). The only exception to this vicinal cysteine pairing is found in dog SP-C, which has one palmitate substituted by a phenylalanine (Fig. 1) (*Johansson et al., 1991*). The palmitoylated groups in SP-C are adjacent to a short polar, N-terminal segment, characterized by cationic residues such as lysine and arginine (Fig. 1). Native SP-C is polymorphic and often has truncations in the N-terminal domain varying from one to three residues. The polar N-terminal region of SP-C (residues ∼1–8) is followed by a highly conserved and hydrophobic C-terminal sequence enriched in polyvaline that spans for ∼26 residues (9–34) to close to the C-terminus (Fig. 1). Prior Fourier transform infrared

(FTIR) spectroscopy showed that native dipalmitoylated SP-C was primarily $\alpha$-helical in surfactant lipid bilayers. Moreover, FTIR spectra indicated that SP-C's long molecular axis was deeply embedded as a transmembrane helix using, hydrogen/deuterium (H/D) exchange and at a slight tilt to the membrane normal with orientation-dependent FTIR analysis (*Vandenbussche et al., 1992*; *Pastrana, Maulone & Mendelsohn, 1991*). Subsequent two-dimensional nuclear magnetic resonance (2D-NMR) of porcine SP-C in a lipid-mimic indicated an unstructured N-terminus (residues ∼1–8), while residues ∼9–34 formed an $\alpha$-helix in the mid- and C-terminal regions (PDB accession code: 1SPF) (*Johansson et al., 1994*). Additional physical studies indicated that the cationic N-terminal domain of SP-C is limited to the surface of phospholipid bilayers and has a higher affinity for anionic than zwitterionic lipids (*Plasencia et al., 2004*). In agreement with these experimental results, recent Molecular Dynamics simulations predicted that SP-C and its attached palmitoyls penetrate deeply into lipid monolayers (*Baoukina & Tieleman, 2010*; *Duncan & Larson, 2010*). MD simulations also confirmed that SP-C embeds into bilayers with its helical axis and palmitoyl groups nearly parallel to the fatty acyl chains and its cationic N-terminal region restricted to the membrane headgroup (*Baoukina & Tieleman, 2010*).

Previous *in vitro* and *in vivo* studies have demonstrated that helical SP-C mediates important surfactant activities in the human lung (*Walther et al., 2007a*; *Johansson, 1998*). With or without the covalent-linked palmitoyl groups, SP-C was shown using various physical techniques to stabilize phospholipid bilayer ensembles that remain attached to the monolayer at the air–water interface. Such adducts may constitute a surfactant lipid 'reservoir' enriched in SP-C that maintains the lipid monolayer throughout the respiration cycle (*von Nahmen et al., 1997*; *Kramer et al., 2000*; *Ding et al., 2001*; *Takamoto et al., 2001*). The presence of SP-C in lung surfactant also enhances the resistance to surfactant flow by increasing surface viscosity (*Alonso & Zasadzinski, 2004*). SP-C plays a lesser role in lung surfactant activities than SP-B. Complete absence of SP-B is lethal in human newborns and animals (*Clark et al., 1995*), but animals with an SP-C mutation are viable with a disease that disrupts lung development (*Bridges et al., 2003*) and increases susceptibility to infection (*Bridges et al., 2006*). Thus, a total synthetic replacement that successfully treats patients with surfactant deficiencies is likely to include mimics of both SP-B and SP-C proteins (*Curstedt, Calkovska & Johansson, 2013*; *Almlen et al., 2010*; *Walther et al., 2014*).

Prior investigations indicate that the two palmitoylated cysteine moieties of SP-C play critical roles in the structural and functional properties of this surfactant protein. These neighboring palmitoyl groups stabilize the $\alpha$-helix of SP-C (*Vandenbussche et al., 1992*; *Pastrana, Maulone & Mendelsohn, 1991*; *Johansson et al., 1994*; *Wang et al., 1996*; *Johansson et al., 1995*; *Dluhy et al., 2003*) and minimize the formation of $\beta$-sheet oligomers (*Johansson, 1998*; *Gustafsson et al., 1999*), optimize surface active interactions with lipids (*Wang et al., 1996*; *Baumgart et al., 2010*), and may induce a coupling between neighboring layers and even trigger a stacking of bilayers (*Baumgart et al., 2010*; *Na Nakorn et al., 2007*). Although SP-C exists primarily as a dipalmitoylated, $\alpha$-helical monomer in bovine lung lavage, there is also a minor dimeric component (∼10%) that is *depalmitoylated* (i.e., lacks the two palmitoyls) and exhibits predominantly intermolecular $\beta$-sheet and abnormal

*in vitro* surfactant activities (*Baatz et al., 1992*). Analogous oligomers of non-palmitoylated SP-C are also connected to pulmonary alveolar proteinosis. *Voss et al. (1992)* earlier reported that partially, or even completely, depalmitoylated SP-C, was present in the bronchoalveolar fluid of patients with pulmonary alveolar proteinosis, and suggested that this pathological modification may reduce surfactant function. This hypothesis is also supported by the diminished *in vitro* surfactant activities seen with chemically deacylated SP-C (*Wang et al., 1996*), and also the high $\beta$-sheet observed with non-palmitoylated SP-C fibrils using FTIR spectroscopy (*Dluhy et al., 2003*). More recently, SP-C, with or without palmitoyls, has been identified in the insoluble aggregates of pulmonary alveolar proteinosis fluid and shows characteristic amyloid properties such as Congo red staining and alveolar fibril formation on electron microscopy (*Gustafsson et al., 1999*). Mutations in the SP-C gene have been associated with familial interstitial lung disease (*Nogee et al., 2001*) and accumulation and misrouting of incorrectly folded proteins (*Whitsett, 2002*). Direct observations of amyloid deposits containing mature SP-C in lung tissue from interstitial lung disease patients with mutations in the BRICHOS domain strongly support this association (*Willander et al., 2012*). Altogether the various structural studies demonstrate that SP-C is metastable in lipid environments and may partially convert from predominately $\alpha$-helix into insoluble and inactive 'amyloid-like' fibrils consisting of $\beta$-sheet.

An early approach to developing SP-C mimics for surfactant replacement therapy was to largely reproduce the sequences of native SP-C using chemically synthesized or recombinant molecules. These *first generation* SP-C mimics primarily follow the native sequence, but typically omit the covalently attached palmitoyl groups to simplify peptide synthesis and purification. For example, the 34-residue SP-Css mimic principally uses the human SP-C sequence as a template but substitutes the palmitoylcysteine groups with serine residues (Fig. 1). The serine replacements approximate the stereochemistry, hydropathy and net charge of native cysteine without being subject to oxidative side-reactions, and thus SP-Css may be a more stable mimic of *depalmitoylated* SP-C. The 34-residue SP-Cff (Fig. 1) also uses human SP-C as a template but the palmitoylcysteine groups are replaced with phenylalanines to simulate *dipalmitoylated* SP-C. Because one of the palmitates in dog SP-C is replaced by phenylalanine (Fig. 1) (*Johansson et al., 1991*), SP-Cff might exhibit high surfactant activity because its two phenylalanines have bulky hydrophobic side chains that may reproduce the insertion of palmitoyls into the hydrophobic interior of surfactant lipid monolayers/bilayers. Unfortunately, research on chemically synthesized, 35-residue SP-Css or SP-Cff indicated diminished *in vitro* surfactant activities in comparison with native SP-C and this may be related to their reduced helical content in lipid environments (*Johansson et al., 1995*). In agreement with these findings, synthetic 34-residue SP-Cff (Fig. 1) showed low solubility in chloroform-methanol, and was difficult to formulate with Tanaka lipids in an active $\alpha$-helical conformation (*Walther et al., 2000*). A range of *in vitro*, *in vivo* and clinical studies have assessed the structural and functional properties of a recombinant, 35-residue SP-Cff (i.e., rSP-C or Venticute® in Fig. 1). Extensive 2D-NMR analyses (*Luy et al., 2004*; *Kairys, Gilson & Luy, 2004*) indicated that freshly prepared recombinant SP-C in chloroform-methanol is a mixture of monomers and dimers,

with a $\alpha$-helical conformation (residues Phe-5 to Leu-34) that largely overlaps that of porcine SP-C (*Johansson et al., 1994*). Non-palmitoylated rSP-C shares key metastable characteristics of native porcine SP-C when suspended in chloroform-methanol for several days, such as the conversion of $\alpha$-helical conformers to insoluble $\beta$-sheet multimers (*Luy et al., 2004*). Although early *in vivo* studies with this recombinant SP-Cff showed surfactant activities closely resembling those of native SP-C (*Ikegami & Jobe, 1998*), a recent clinical trial of patients with ARDS demonstrated that rSP-C surfactant neither decreased mortality nor improved patient oxygenation (*Spragg et al., 2011*). Pulsating bubble surfactometry results suggested that this may have been due to a partial surfactant inactivation during resuspension of rSP-C prior to administration (*Spragg et al., 2011*). Consequently, the low or unstable surfactant activities observed for the above SP-Css and SP-Cff peptides (Fig. 1) argue that further sequence changes are required to produce more effective, non-palmitoylated SP-C mimics.

Accordingly, a *second generation* of SP-C mimics lacking palmitoyl groups, but with enhanced $\alpha$-helical content and surfactant activities through amino-acid substitutions in the primary sequence, is now being developed for surfactant replacement therapy. The chemically synthesized SP-C33 (Fig. 1) is a second generation SP-C mimic, in which the adjacent palmitoylcysteine groups are replaced with serines and the poly-valines in the $\alpha$-helix are substituted with multiple leucines. *In vitro* and *in vivo* studies demonstrated surfactant activities for this SP-C33 analog comparable to those of native SP-C (*Johansson et al., 2003*; *Almlen et al., 2011*) and this may be because SP-C33 and native SP-C both similarly insert into surfactant lipids as slightly tilted, transmembrane $\alpha$-helices (*Vandenbussche et al., 1992*; *Pastrana, Maulone & Mendelsohn, 1991*; *Almlen et al., 2011*). These results support the hypothesis that the manipulation of native sequences may produce SP-C mimics with enhanced $\alpha$-helicity and surfactant activity. However, SP-C33 may not be completely optimized for such important characteristics as surfactant activity (alone or in combination with other surfactant components such as SP-B mimics), resistance to inhibition, toxicity or shelf-life stability. Less invasive second generation SP-C mimics using fewer substitutions may permit more accurate control over critical surfactant properties.

Another strategy for obtaining second generation SP-C mimics without palmitoyl groups involves the introduction of charged ion-pairs (i.e., 'salt-bridges' or 'ion-locks') into SP-C peptides, in which the cationic $Lys^+$ or $Arg^+$ residues interact with anionic $Glu^-$ or $Asp^-$ residues. In Circular Dichroism (CD) experiments, *Marqusee & Baldwin (1987)* showed that incorporation of $Lys^+$ and $Glu^-$ at intervals of 4 residues (i.e., "$i + 4$") in the sequence substantially increased the helix content of the host peptide. This stabilization of the helix in aqueous environments is due to the formation of an electrostatically neutral ion-pair *via* the positive- and negative-charged side groups for the $Lys^+$ and $Glu^-$ residues. Using an augmented Wimley-White (WW) hydrophobicity scale, which accounts for whole-residue energy contributions by the peptide backbone, side-chains and salt-bridge pairs, Membrane Protein Explorer (MPEx) analyses predicted that 'ion-locks' such as $Glu^- – Lys^+$ may increase the helicity and lipid-partitioning of peptides and proteins in membranes (*Snider et al., 2009*). Because the formation of an ion-pair is electrostatically

neutral, MPEx showed that there is only a modest thermodynamic penalty for replacing the hydrophobic residues of a transmembrane sequence with an 'ion-lock' pairing (e.g., Glu$^-$ and Lys$^+$) (*Jayasinghe, Hristova & White, 2001*). Salt-bridges (or 'ion-locks') have been found in numerous proteins such as Ca$^{2+}$-ATPase, which has a membrane helix stabilized by an intrahelical charge-pair when facing lipids (*Bano-Polo et al., 2012*). Importantly, electrostatic calculations indicated that salt-bridges with close-range distances ($\leq 4.0$ Å) between the cationic residue "N" and the nearest anionic residue "O" are mostly $\alpha$-helix stabilizing, while those with long-range distances ($>5$ Å) may be helix destabilizing (*Kumar & Nussinov, 1999*; *Kumar & Nussinov, 2002*).

In the present study, we synthesized peptides representing the native sequence of SP-C (SP-Css and SP-Cff) lacking palmitoyl groups, an SP-C mimic with elevated surfactant activity (SP-C33 UCLA), and SP-C mimics with single or double salt-bridges in the mid-section of the hydrophobic C-terminal region (i.e., SP-Css ion-lock 1, SP-Cff ion-lock 1, SP-C33 ion-lock 1 and SP-C33 ion-lock 2; see Fig. 1). FTIR spectroscopy of SP-Css and SP-Cff in surfactant lipids, an aqueous buffer or a $\alpha$-helix promoting HFIP buffer (*Crescenzi et al., 2002*; *Rajan et al., 1997*) indicated elevated $\beta$-sheet, while the corresponding spectra of the daughter SP-Css ion-lock 1 and SP-Cff ion-lock 1 in these environments demonstrated low $\beta$-sheet and mostly high $\alpha$-helix. However, FTIR spectroscopy also indicated that the $\alpha$-helicity of SP-Css ion-lock 1 in surfactant lipids was higher than that of SP-Cff ion-lock 1. Furthermore, oriented and hydrogen/deuterium (H/D) exchange FTIR spectra showed that an SP-Css ion-lock 1 incorporates into lipid bilayers as a slightly tilted transmembrane $\alpha$-helix, while SP-Cff ion-lock 1 with its shorter helix had a greater tilt and may be restricted to only one monolayer. Consequently, the much higher *in vitro* and *in vivo* surfactant activities reported here for SP-Css ion-lock than for SP-Cff ion-lock 1 are attributed to the former SP-C mimic more closely simulating the enhanced $\alpha$-helicity and membrane topography of the native dipalmitoylated SP-C. These results provide a proof of concept that selective insertion of salt-bridge(s) may produce stable and functional SP-C mimics that completely block the well-known amyloidogenic propensities of the parent SP-C protein.

## MATERIALS & METHODS

### Ethics statement

The animal study was reviewed and approved by the Institutional Animal Care and Use Committee of the Los Angeles Biomedical Research Institute at Harbor-UCLA Medical Center (protocol # 12958). All procedures and anesthesia were performed in accordance with the American Veterinary Medical Association guidelines.

### Materials

All organic solvents for sample synthesis were HPLC grade or better. Phospholipids were obtained from Avanti Polar Lipids, Inc. (Alabaster, AL). Deuterated water was NMR quality from Aldrich Chemical Co (St. Louis, MO).

## Synthesis of SP-C peptides

A suite of SP-C peptides, composed of the 34-residue SP-Cff, SP-Cff ion-lock 1, SP-Css and SP-Css ion-lock 1 and the 33-residue SP-C33 UCLA, SP-C33 ion-lock 1 and SP-C33 ion-lock 2 (Fig. 1), was synthesized with FastMoc[TM] (*Fields et al., 1991*) or standard Fmoc procedures on a Leu-HMPB NOVA resin using ABI 431A (Applied Biosystems, Foster City, CA), Symphony Multiple Peptide (Protein Technologies, Tucson, AZ) or Liberty Microwave Peptide (CEM Corp, Matthews, NC) synthesizers. The SP-C mimic peptides were each carboxylated at the C-terminus and amidated at the N-terminus, and all residues were double-coupled to insure optimal yield (*Waring et al., 2005*). C-terminal carboxyl SP-C peptides were cleaved from the resin using the standard phenol:thioanisole:ethanedithiol water:trifluoracetic acid (0.75:0.25:0.5:0.5: 10, v:v) cleavage-deprotection mixture (*Walther et al., 2010*). Crude peptides were then purified (better than 95%) by preparative HPLC using a VYDAC diphenyl or C8 (1″ by 12″ width by length) column at 20 ml/min. SP-C peptides were each eluted from the column with a 0–100% (water to acetonitrile with 0.1% TFA as an ion pairing agent added to both aqueous and organic phases) linear gradient in one hour. The purified product was freeze-dried directly and the mass was confirmed by Maldi TOF mass spectrometry.

## Attenuated total reflectance-fourier transform infrared (ATR-FTIR) spectroscopy

Infrared spectra were recorded at 37 °C for selected SP-C peptides in Fig. 1 using a Bruker Vector 22 FTIR spectrometer (Pike Technologies, Madison, WI) with a deuterium triglyceride sulfate (DTGS) detector, averaged over 256 scans at a gain of 4 and a resolution of 2 cm$^{-1}$ (*Waring et al., 2005*; *Walther et al., 2007b*; *Walther et al., 2010*; *Gordon et al., 2008*). For spectral recordings of SP-C peptides in hexafluoroisopropanol (HFIP) or aqueous buffer solutions, self-films were first prepared by air-drying peptide originally in 100% HFIP onto a 50 × 20 × 2 mm, 45° germanium attenuated total reflectance (ATR) crystal for the Bruker spectrometer. The dried peptide self-films were then overlaid with solutions containing 35% HFIP/65% deuterated-10 mM sodium phosphate buffer (pD 7.4) (*Glasoe & Long, 1960*) or 100% deuterated-10 mM sodium phosphate buffer (pD 7.4) before spectral acquisition; control solvent samples were similarly prepared, but without peptide. Spectra of peptides in solvent were obtained by subtraction of the solvent spectrum from that of the peptide-solvent. For FTIR spectra of SP-C peptides in synthetic surfactant lipid multilayers (DPPC:POPC:POPG; weight ratio, 5:3:2), the peptide was co-solvated with lipid systems in HFIP (lipid:peptide mole ratio of 10:1) and transferred onto an ATR crystal (Pike Technologies). The organic solvent was then removed by flowing nitrogen gas over the sample to produce a lipid-peptide film. The lipid-peptide films were next hydrated (≥35%) (*Yamaguchi et al., 2001*) with 10 mM deuterated sodium phosphate buffer (pD 7.4) vapor in nitrogen for 1 h prior to acquiring spectra (*Gordon et al., 2000*; *Yamaguchi et al., 2002*). The spectra for peptide in lipid mixtures were obtained by subtracting the lipid spectrum hydrated with 10 mM deuterated phosphate buffer (pD 7.4) from that of peptide in lipid with 10 mM phosphate buffer (pD 7.4). Infrared spectra were

subsequently recorded for the lipid-peptide films at 37 °C, similar to those described for peptide self-films (see above). Relative amounts of $\alpha$-helix, $\beta$-turn, $\beta$-sheet, or random (disordered) structures in either peptide self-films or lipid-peptide films were estimated using Fourier deconvolution (GRAMS/AI8, version 8.0, Themo Electron Corporation, Waltham, MA) and area of component peaks calculated using curve-fitting software (Igor Pro, version 1.6, Wavemetrics, Lake Oswego, OR) (*Kauppinen et al., 1981*). FTIR frequency limits were: $\alpha$-helix (1662–1645 cm$^{-1}$), $\beta$-sheet (1637–1613 cm$^{-1}$ and 1710–1682 cm$^{-1}$), turn/bend (1682–1662 cm$^{-1}$), and disordered or random (1650–1637 cm$^{-1}$) (*Byler & Susi, 1986*).

To estimate the orientation of the $\alpha$-helix of various SP-C peptides in surfactant lipid multilayers deposited on the germanium crystal surface, gold wire polarizers (Perkin Elmer, Waltham, MA) were rotated from 0° to 90° to obtain polarized IR spectra of each lipid-peptide film (*Gordon et al., 1996*). The insertion (or tilt) angle $\Theta$ for the SP-C helical axis with respect to the normal of the germanium surface was calculated based on the dichroic ratio $R$ (*Beevers & Kukol, 2006*), which is equal to the ratio of $A_{parallel}$ to $A_{perpendicular}$. Here, $A_{parallel}$ and $A_{perpendicular}$ are defined as the respective areas of absorbance at 0° (parallel) and 90° (perpendicular) polarizations for the amide I band centered at ~1656 cm$^{-1}$. These measures assumed a thick film approximation with the values for the electric field components of the evanescent wave to be $E_x = 1.398$, $E_y = 1.516$, $E_z = 1.625$, and an angle $\alpha = 39°$ for the vibrational dipole relative to the molecular axis of the helix to derive an order parameter $S$ (*Beevers & Kukol, 2006*; *Chia et al., 2002*). Here, $S$ is equal to $[(E_x^2 - RE_y^2 + E_z^2)/(E_x^2 - RE_y^2 + E_z^2)] \div [(3\cos^2\alpha - 1)/2]$. The experimental tilt angle, $\Theta$, was then calculated from this order parameter, by noting that $S = (3\cos^2\Theta - 1)/2$. However, it should be noted that the multilayer peptide-lipid samples stacked on the germanium crystal are unlikely to be perfectly ordered (*Arkin, MacKenzie & Brunger, 1997*; *Kukol et al., 1999*), and this introduces an isotropic contribution that will artificially reduce the dichroic ratio $R$. Accordingly, the experimental $\Theta$ angles obtained here with respect to the germanium crystal should be viewed as maximum tilt angles, with the actual tilt angles with respect to the normal of the multilayer lipid surface likely to be lower (*Beevers & Kukol, 2006*).

For selected SP-C peptides in surfactant lipids, the stability of the $\alpha$-helical component was also analyzed by monitoring the hydrogen/deuterium (H/D) exchange for amide protons with the FTIR spectrometer. The peptide-lipid sample on the ATR crystal was flushed with 10 mM deuterated phosphate buffer (pD 7.4) vapor in nitrogen at 37 °C, and the decay of the amide II band area (1525–1565 cm$^{-1}$) was determined as a function of time (0–6 h) to assess the time-dependent H/D exchange of the amide group (*Vandenbussche et al., 1992*; *Beevers & Kukol, 2006*; *Almlen et al., 2011*).

### Prediction of aggregation-forming regions in SP-C peptides

SP-C peptides were analyzed with PASTA (*Trovato et al., 2006*; *Trovato, Seno & Tosatto, 2007*) to characterize those domains most likely to form $\beta$-sheet, particularly in polar environments such as aqueous buffer or the lipid–water interface of membranes

(*Gordon et al., 2008*). The PASTA algorithm systematically calculates the relative energies of the various pairing arrangements by assessing a pair-wise energy function for residues facing one another within a $\beta$-sheet. Using a database of known 3D-native structures, PASTA computes two different propensity sets depending on the directionality (i.e., parallel or antiparallel $\beta$-sheets) of the neighboring strands. PASTA assigns relative energies to specific $\beta$-pairings of two sequence stretches of the same length, and assumes that the lower relative energies enhance aggregation by further stabilizing the cross-$\beta$ core. Theoretical PASTA predictions of SP-C and SP-C mimic aggregation were performed by submitting the primary sequences of these peptides (Fig. 1) to the PASTA Version 1.1 (http://protein.cribi.unipd.it/pasta) website (*Trovato, Seno & Tosatto, 2007*). PASTA calculates the relative energies of the most stable pairings (i.e., more negative energies denote more stable conformations), and outputs them as an HTML file (maximum pairings: 100). The PASTA energies for the top-ranked pairings may predict whether the $\beta$-sheet domains of one peptide will be more stable than those of other peptides. The energies for these pairings may also be used to compute an aggregation profile for those regions within a peptide most likely to aggregate as $\beta$-sheets. One cautionary note in using the PASTA algorithm is that it does not correct for the actual environmental polarity of a given protein domain, and also does not explicitly account for salt-bridges forming between charged ion-pairs.

## Hydropathy predictions of SP-C peptides inserting into lipid bilayers as transmembrane helices

Membrane Protein Explorer (MPEx; Version 3.2.9) is a Java program that analyses hydrophobic lipid–protein interactions in membranes (http://blanco.biomol.uci.edu/mpex). With the hydropathy analysis mode (*Snider et al., 2009*), hydropathy plots are produced using the augmented Wimley-White (WW) whole-residue hydrophobicity scale that predicts transmembrane helices for protein sequences with high accuracy (*Jayasinghe, Hristova & White, 2001*; *Liang et al., 2012*). This augmented scale is experimentally based on the partitioning of hydrophobic pentapeptides (*Wimley, Creamer & White, 1996*) and salt-bridge pairs (*Wimley et al., 1996*) into $n$-octanol (i.e., solvent mimic of the hydrocarbon (non-polar) region of the bilayer), and accounts for whole-residue energy contributions due to the peptide backbone, side-chains and salt-bridge pairs (*Jayasinghe, Hristova & White, 2001*; *Snider et al., 2009*). Protein sequences may be submitted to the MPEx program, and the resulting plots are presented as hydropathy (kcal/mol) versus the sequence residue number, averaged over a sliding window of 19 amino-acid residues. Higher positive hydropathy values reflect enhanced lipid bilayer partitioning for any putative membrane helices. The default scale for the hydropathy analysis uses $n$-octanol, and there is an option for engaging potential salt-bridges using an "on–off" switch on the MPEx control panel.

### *In vitro* adsorption surface pressure measurements

Adsorption experiments were done at $37 \pm 0.5\,°C$ in a Teflon® dish with a 35 ml subphase ($0.15$ M NaCl $+ 1.5$ mM $CaCl_2$) stirred to minimize diffusion resistance as described

previously (*Notter, Finkelstein & Taubold, 1983*). Surfactants were prepared by co-solvating the lipid components (i.e., DPPC:POPC:POPG, 5:3:2, weight ratio) with the SP-C peptide in organic solvent HFIP (100%), dried under nitrogen, exposed to house vacuum to remove residual solvent, resuspended by hand vortexing in 0.15 M NaCl (normal saline) adjusted to pH 7.0 with 0.1 N sodium bicarbonate, heated to 65 °C intermittently for 30 min, and refrigerated for 12 h prior to use. At time zero, a bolus of these surfactant preparations containing 2.5 mg lipid in 5 ml of 0.15 M NaCl + 1.5 mM $CaCl_2$ was injected into the stirred subphase, and adsorption surface pressure (surface tension lowering below that of the pure subphase) was measured as a function of time by the force on a partially submerged, sandblasted platinum Wilhelmy slide (*Notter, Finkelstein & Taubold, 1983*). The final surfactant concentration for adsorption studies was uniform at 0.0625 mg phospholipid/ml (2.5 mg surfactant phospholipid/40 ml of final subphase).

### *In vitro* surfactant activity measured with captive bubble surfactometry

Adsorption and dynamic surface tension lowering ability of all above described surfactant preparations were measured with a captive bubble surfactometer at physiological cycling rate, area compression, temperature, and humidity (*Walther et al., 2010*). We routinely analyze surfactant samples of 1 µl (35 mg phospholipids/ml) in the captive bubble surfactometer and perform all measurements in quadruplicate.

### Ventilated, lung-lavaged, surfactant-deficient rabbit model

Young adult New Zealand white rabbits (weight 1.0–1.3 kg) received anesthesia with 50 mg/kg of ketamine and 5 mg/kg of acepromazine intramuscularly prior to placement of a venous line via a marginal ear vein. After intravenous administration of 1 mg/kg of diazepam and 0.2 mg/kg of propofol, a small incision was made in the skin of the anterior neck for placement of an endotracheal tube and a carotid arterial line. After placement of the endotracheal tube muscle paralysis was induced with intravenous pancuronium (0.1 mg/kg). During the ensuing duration of mechanical ventilation anesthesia was maintained by continuous intravenous administration of 3 mg/kg/h of propofol and intravenous dosages of 1 mg/kg of diazepam as needed; muscle paralysis was maintained by hourly intravenous administration of 0.1 mg/kg of pancuronium. Heart rate, arterial blood pressures and rectal temperature were monitored continuously. Respiratory function was followed during mechanical ventilation by measurements of serial arterial blood gases and pulmonary compliance over the first 2 h after a standardized saline lavage protocol. For this timeframe of study, the model reflected a relatively pure state of surfactant insufficiency in animals with mature lungs.

After stabilization on the ventilator, saline lavage was performed with repeated intratracheal instillation and removal of 30 mL of normal saline until the $PaO_2$ dropped below 100 mm Hg (average 3 lavages). Maintenance fluid was provided by a continuous infusion of Lactated Ringer's solution at a rate of 10 ml/kg/h. Edema fluid appearing in the trachea was removed by suctioning. When the $PaO_2$ was stable at less than 100 mm Hg in saline lavaged animals, an experimental or control surfactant mixture was instilled

into the trachea at a dose of 100 mg/kg body weight and a concentration of 35 mg/ml. The rabbits were ventilated using a Harvard volume controlled animal ventilator (tidal volume 7.5 ml/kg, a positive end-expiratory pressure of 3 cm $H_2O$, an inspiratory/expiratory ratio of 1:2, 100% oxygen, and a respiratory rate to maintain a $PaCO_2$ at ~40 mm Hg). Airway flow and pressures and tidal volume were monitored continuously with a pneumotachograph connected to the endotracheal tube and a pneumotach system. Arterial pH and blood gases were done every 15 min. Dynamic lung compliance was calculated by dividing tidal volume/kg body weight by changes in airway pressure (peak inspiratory pressure minus positive end-expiratory pressure) (ml/kg/cm $H_2O$). Animals were sacrificed 2 h after surfactant administration with an overdose of pentobarbital. End-points were pulmonary gas exchange (arterial pH, $PaCO_2$ and $PaO_2$) and pulmonary mechanics (dynamic compliance).

## RESULTS

### FTIR spectroscopic analysis of the secondary conformations for SP-C mimics with and without ion-locks

To test the structural effects of inserting ion-locks into SP-C peptides, a suite of SP-C mimics was prepared based on native SP-C sequences (i.e., SP-Css and SP-Cff) and these native SP-C sequences with a $Glu^-$–$Lys^+$ pairing in the hydrophobic midsection (i.e., SP-Css ion-lock 1 and SP-Cff ion-lock 1; see Fig. 1). Representative FTIR spectra of the amide I band for SP-Css and SP-Cff in surfactant lipids (DPPC:POPC:POPG 5:3:2, weight ratio) in Fig. 2A were similar, each showing a major $\beta$-sheet component with a predominant peak at ~1630–1631 $cm^{-1}$ and a miniscule companion peak at ~1691 $cm^{-1}$, and a lower $\alpha$-helical component with a mid-field shoulder at ~1657 $cm^{-1}$. Subsequent deconvolutions confirmed that SP-Css and SP-Cff in surfactant lipids principally adopt $\beta$-sheet conformations ($\geq$~58%) with minor contributions from $\alpha$-helix, loop-turn and disordered structures (Table 1). In additional studies (Table 2), FTIR spectra and deconvolutions for SP-Css and SP-Cff in a membrane-interfacial lipid mimic (i.e., 35% HFIP/65% deuterated-sodium phosphate buffer, pD 7.4) or 100% aqueous buffer (i.e., 10 mM deuterated phosphate buffer, pD 7.4) indicated comparably high $\beta$-sheet ($\geq$36%–50%), with smaller proportions from other conformations. Consistent with these findings, prior FTIR spectroscopy of 34-residue SP-Cff showed reduced $\alpha$-helix on short-term storage in surfactant lipids (*Walther et al., 2000*), while earlier CD analyses of synthetic, 35-residue SP-Css or SP-Cff in detergent micelles also indicated enhanced $\beta$-sheet and reduced $\alpha$-helix (<20%) (*Johansson et al., 1995*). In our hands, SP-Css and SP-Cff exhibited only high $\beta$-sheet and low $\alpha$-helix in surfactant lipids (Fig. 2; Table 1) or environments of differing polarity (Table 2).

FTIR spectroscopy determinations for either SP-Css ion-lock 1 or SP-Cff ion-lock 1 in surfactant lipids indicated enhanced $\alpha$-helix and reduced $\beta$-sheet, when compared to the corresponding measurements of the native SP-Css or SP-Cff sequences. For example, FTIR spectra of SP-Css ion-lock 1 in surfactant lipids (Fig. 2A) indicated high $\alpha$-helix (at ~1657 $cm^{-1}$), and very low $\beta$-sheet (at ~1630 $cm^{-1}$). Importantly, deconvolution
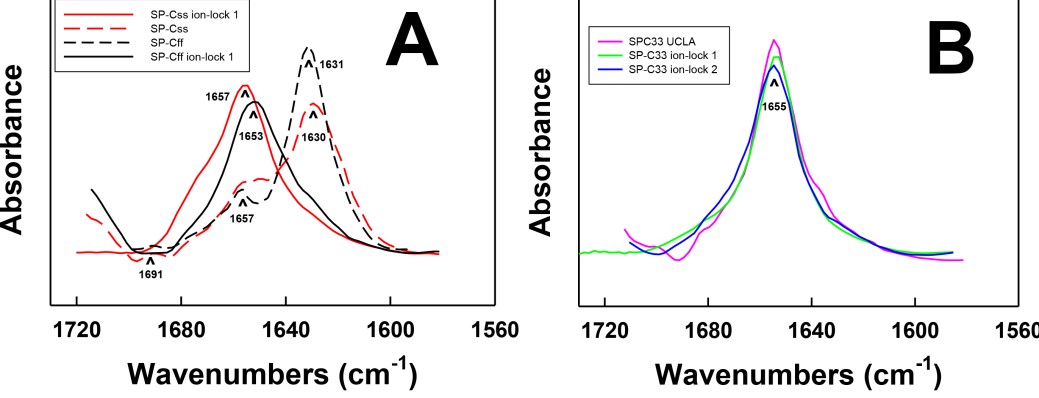

**Figure 2 FTIR spectra of the amide I bands for SP-C mimic peptides in surfactant lipids (DPPC:POPC:POPG 5:3:2).** (A) SP-Css is the dashed red line, SP-Css ion-lock 1 is the solid red-line, SP-Cff is the dashed black line and SP-Cff ion-lock 1 is the solid black line. The spectra for either SP-Css or SP-Cff in surfactant lipids with 10 deuterated mM phosphate buffer (pD 7.4) principally show $\beta$-sheet, with a major peak at 1630–1631 cm$^{-1}$ and a very low companion peak at $\sim$1691 cm$^{-1}$. The SP-Css and SP-Cff spectra also show significant shoulders at $\sim$1653–1657 cm$^{-1}$, indicating minor $\alpha$-helical structures. The corresponding spectra for SP-Css ion-lock 1 and SP-Cff ion-lock 1 in surfactant lipids primarily show $\alpha$-helix with maxima at $\sim$1653–1657 cm$^{-1}$, indicating that incorporation of the Glu$^-$–Lys$^+$ ion-pair in either SP-C mimic effectively reverses their $\beta$-sheet properties. (B) SP-C33 UCLA is a solid magenta line, SP-C33 ion-lock 1 is a solid green line, and SP-C33 ion-lock 2 is a solid blue line. The spectra for SP-C33 UCLA and the SP-C33 ion-locks in lipids all closely overlap, with shared maxima at $\sim$1655 cm$^{-1}$ indicating predominate $\alpha$-helix. The introduction of a single or double Glu$^-$–Lys$^+$ ion-pair in SP-C33 does not raise the $\alpha$-helical levels over that seen with the parent SP-C33 UCLA (Table 1). Peptide concentrations were 470 µM, and sequences are in Fig. 1. FTIR spectra for peptide in lipid mixtures (Figs. 2A and 2B) were obtained by subtracting the lipid spectrum hydrated with 10 mM deuterated phosphate buffer (pD 7.4) from those of peptides in lipid with 10 mM phosphate buffer (pD 7.4).

**Table 1** Proportions of secondary structure[a] and maximum tilt angles ($\Theta$) of the membrane $\alpha$-helix[b] for SP-C peptide mimics in synthetic surfactant lipids[c] with 10 mM deuterated phosphate buffer (pD 7.4), as estimated from the ATR-FTIR spectra of the peptide amide I band.

| System | % Conformation[a] | | | | Max. tilt angle[b] |
|---|---|---|---|---|---|
| | $\alpha$-helix | $\beta$-sheet | Loop-turn | Disordered | $\theta$ |
| *DPPC:POPC:POPG[c]* | | | | | |
| SP-Cff | 5.6 | 74.3 | 2.9 | 17.2 | — |
| SP-Cff ion-lock 1 | 44.6 | 17.8 | 15.7 | 22.0 | 34.2° |
| SP-Css | 24.5 | 58.6 | 9.1 | 7.9 | — |
| SP-Css ion-lock 1 | 63.1 | 5.5 | 25.5 | 5.9 | 22.6° |
| SP-C33 | 67.2 | 6.9 | 13.9 | 12.0 | 22.5° |
| SP-C33 ion-lock 1 | 66.9 | 3.6 | 21.7 | 7.8 | 22.7° |
| SP-C33 ion-lock 2 | 65.1 | 7.7 | 18.0 | 9.18 | 23.2° |

**Table 2  Proportions of secondary structure for SP-C peptide mimics in deuterated phosphate and hexafluoroisopropanol buffer.** Proportions of secondary structure[a] for SP-C peptide mimics in 10 mM deuterated phosphate buffer (dPBS, pD 7.4) and 35% hexafluoroisopropanol (HFIP)/65% dPBS, as estimated from the ATR-FTIR spectra of the peptide amide I band.

| System | % Conformation[a] | | | |
| --- | --- | --- | --- | --- |
| | $\alpha$-helix | $\beta$-sheet | Loop-turn | Disordered |
| *dPBS* | | | | |
| SP-Cff | 7.4 | 51.9 | 24.0 | 10.6 |
| SP-Cff ion-lock 1 | 17.8 | 2.9 | 59.9 | 19.5 |
| SP-Css | 15.8 | 49.7 | 14.6 | 19.9 |
| SP-Css ion-lock 1 | 40.8 | 11.8 | 18.8 | 28.7 |
| SP-C33 UCLA | 50.0 | 14.4 | 17.2 | 18.4 |
| SP-C33 ion-lock 1 | 57.8 | 14.8 | 12.2 | 15.1 |
| *35% HFIP/dPBS* | | | | |
| SP-Cff | 19.3 | 53.2 | 17.8 | 9.7 |
| SP-Cff ion-lock 1 | 53.0 | 13.9 | 16.9 | 16.1 |
| SP-Css | 18.5 | 35.6 | 22.1 | 23.9 |
| SP-Css ion-lock 1 | 51.1 | 10.4 | 33.1 | 5.45 |
| SP-C33 UCLA | 52.6 | 14.1 | 18.6 | 14.7 |
| SP-C33 ion-lock 1 | 49.6 | 8.01 | 1.42 | 13.7 |

showed predominant $\alpha$-helix (i.e., $\sim$63%) and minor $\beta$-sheet ($\sim$6%) for SP-Css ion-lock 1 in surfactant lipids, as opposed to the reduced $\alpha$-helix ($\sim$25%) and elevated $\beta$-sheet ($\sim$58%) determined from FTIR spectra of its parent SP-Css in this environment (Fig. 2A; Table 1). Consequently, insertion of the ion-lock (i.e., Glu$^-$-20–Lys$^+$-24) in the mid-section of the poly-Val sequence produced a remarkable conformation shift in the host SP-Css, from an 'amyloid-like' $\beta$-sheet profile to one that is primarily $\alpha$-helix (Fig. 2A). Additional FTIR spectra of SP-Css ion-lock 1 in aqueous and membrane-interfacial HFIP milieu also demonstrated elevated $\alpha$-helix values ($\geq \sim$41%) over those obtained for the parent SP-Css (Table 2), suggesting that an intrapeptide salt-bridge (at Glu$^-$-20–Lys$^+$-24) stabilizes the $\alpha$-helical conformation in environments of differing polarity. In further studies, FTIR spectra of SP-Cff ion-lock 1 in surfactant lipids showed enhanced $\alpha$-helix (at $\sim$1653 cm$^{-1}$) and low $\beta$-sheet (at $\sim$1630 cm$^{-1}$) (Fig. 2A). Deconvolution of these SP-Cff ion-lock 1 spectra verified that the major structural component in lipids was $\alpha$-helix ($\sim$44.6%), produced by a conformational change from the very high $\beta$-sheet ($\sim$74.3%) of its parent SP-Cff (Fig. 2A; Table 1). Incorporation of the ion-pair (i.e., Glu$^-$-20–Lys$^+$-24) induced a similar conformational shift for SP-Cff in the lipid-mimic HFIP solution from a $\beta$-sheet conformation ($\sim$53.2%) to one that is predominately $\alpha$-helix ($\sim$53.0%) (Table 2). Although these FTIR spectral results suggest at least a partial salt-bridge for SP-Cff ion-lock 1 in lipid and lipid-mimic environments, additional FTIR experiments indicate that this salt-bridge may not be as stable as that seen with SP-Css ion-lock 1. Unlike SP-Css ion-lock 1, SP-Cff ion-lock 1 folds primarily as loop-turns in aqueous buffer (Table 2) and even shows diminished $\alpha$-helix content in surfactant lipids relative to that of SP-Css ion-lock 1

(i.e., ~44.6 vs. ~63.1% in Fig. 2A and Table 1). Such $\alpha$-helical reductions may be partially due to the Phe groups of SP-Cff ion-lock 1 (Fig. 1) weakening the putative salt-bridge between Glu$^-$-20–Lys$^+$-24, either through direct or indirect mechanisms (see below).

It is also worthwhile to contrast the helical properties determined for FTIR spectra of SP-Css ion-lock 1 and SP-Cff ion-lock 1 with those of SP-C33 UCLA and several SP-C33 ion-locks. SP-C33 UCLA has the same sequence as SP-C33 (*Almlen et al., 2010*; *Almlen et al., 2011*) with poly-Leu substituting for the poly-Val in the C-terminal region (Fig. 1). Previous CD and FTIR studies have reported that SP-C mimics of the poly-Leu class, including SP-C33 (*Almlen et al., 2011*) and its homologous precursor SP-C(Leu) (*Nilsson et al., 1998*), each show elevated $\alpha$-helix when incorporated into lipids or lipid-mimics. In support of these experimental results, earlier Molecular Dynamics simulations of SP-C33 in methanol for 10 ns indicated that the hydrophobic poly-Leu region adopted an $\alpha$-helical conformation in this membrane-interfacial mimic (*Almlen et al., 2010*). The FTIR spectrum of SP-C33 UCLA in surfactant lipids overlapped that of SP-Css ion-lock 1 (Figs. 2A and 2B), and spectral deconvolution indicated similarly high $\alpha$-helix (Table 1). Moreover, FTIR spectra of SP-C33 UCLA in either aqueous or HFIP solutions showed $\alpha$-helical components (at ~1653–1655 cm$^{-1}$) nearly identical to those of SP-Css ion-lock 1 (*not shown*), as well as similar $\alpha$-helix levels (i.e., ~40%–51%) using spectral deconvolution (Table 2). Although SP-Css ion-lock 1 and SP-C33 UCLA indicate comparably high $\alpha$-helix in aqueous buffer, aqueous HFIP buffer or surfactant lipids (Fig. 2: Tables 1 and 2), SP-C33 UCLA requires replacement of the poly-Val sequence with ten Leu residues to achieve this stabilization. Contrarily, SP-Css ion-lock 1 needs only a single ion-pair at Glu$^-$-20–Lys$^+$-24 to maintain high $\alpha$-helix. In further research, the SP-C33 ion-lock 1 and SP-C33 ion-lock 2 mimics were prepared by inserting a single (i.e., Glu$^-$-20–Lys$^+$-24) or double (i.e., Lys$^+$-15–Glu$^-$-19 and Glu$^-$-20–Lys$^+$-24) ion-lock into the SP-C33 sequence (Fig. 1). FTIR spectra and deconvolution indicated $\alpha$-helix for these SP-C33 ion-locks in aqueous, HFIP-water and surfactant lipids similar to those of SP-C33 (Fig. 2; Tables 1 and 2). For example, the FTIR spectra of SP-C33 UCLA and the SP-C33 ion-locks in surfactant lipids nearly overlap with $\alpha$-helix maxima at 1655 cm$^{-1}$ (Fig. 2B), and spectral deconvolutions of each show elevated $\alpha$-helix of ~65%–67% (Table 1). Because the insertions of either one or two charged ion-pairs were unable to raise the $\alpha$-helical content beyond that of the parent SP-C33 UCLA (Tables 1 and 2), the SP-C33 primary sequence may already be maximized for $\alpha$-helix. Interestingly, the $\alpha$-helicity of SP-Css ion-lock 1 in surfactant lipids (i.e., ~63%) is only slightly below those of the SP-C33 and the SP-C33 ion-locks (i.e., ~65%–67%), suggesting that the SP-Css ion-lock 1 in lipids is also almost optimized for $\alpha$-helix.

## Effects of ion-locks on the aggregation domains of SP-C peptides detected with PASTA analysis

PASTA calculations were next performed to assess whether the enhanced $\alpha$-helicity induced by ion-lock insertion into either SP-Css or SP-Cff (Fig. 2; Tables 1 and 2) is related to the disruption of $\beta$-sheet regions in the host peptides. The PASTA algorithm systematically

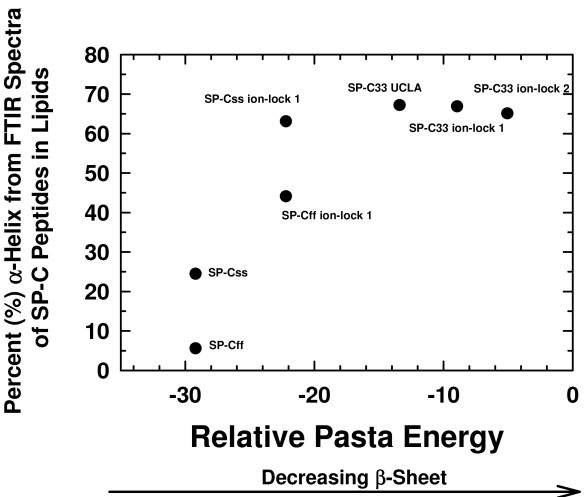

**Figure 3  Plot of α-helix (%) from FTIR spectra of SP-C peptides in surfactant lipids versus relative PASTA energies.** Percent (%) α-helix was determined from the FTIR spectra of SP-Css, SP-Cff, SP-Css ion-lock 1, SP-Cff ion-lock 1, SP-C33 UCLA, SP-C33 ion-lock 1 and SP-C33 ion-lock 2 (Fig. 2; Table 1). Relative energy values were from PASTA, with less negative values reflecting a lower propensity to form β-sheet. The PASTA energies for SP-Css, SP-Cff, SP-Css ion-lock 1 and SP-Cff ion-lock 1 are calculated from the C-terminal residues 7–27, while those from SP-C33, SP-C33 ion-lock 1 and SP-C33 ion-lock 2 are from the C-terminal segment 6–26. The sequences and numbering for the SP-C peptides are in Fig. 1.

determines the relative energies for various paired sequences by assessing an energy function for facing residues that are H-bonded within a β-sheet (*Trovato et al., 2006*). Here, PASTA predicted that the native SP-Cff and SP-Css peptides will each form β-sheet at their respective C-terminal domains. Specifically, PASTA energy calculations showed that the most likely pairing for SP-Cff or SP-Css will be an interpeptide, parallel β-sheet (i.e., parallel in-register arrangement (PIRA)) for the C-terminal residues 7–27 (Fig. 1) with a relative PASTA energy of −29.2 (*Trovato et al., 2006*; *Trovato, Seno & Tosatto, 2007*). Using a database of 179 peptides as a benchmark (*Trovato, Seno & Tosatto, 2007*), an energy threshold of $< -29.0$ indicates a probability $> 98\%$ (true positive rate) that SP-Cff and SP-Css are each "amyloid-like" (i.e., have a high β-sheet tendency) for the corresponding C-terminal segments (i.e., residues 7–27). This prediction of elevated β-sheet for SP-Css and SP-Cff was confirmed by FTIR spectra of these peptides in surfactant lipids with high β-sheet and low α-helix (Figs. 2 and 3; Table 1). Furthermore, Fig. 3 shows that incorporation of Glu$^-$-20–Lys$^+$-24 in either SP-Cff or SP-Css increased the relative PASTA energy from −29.2 to −22.2 for the 7–27 sequence, and also raised the α-helix levels of the particular SP-C ion-locks in surfactant lipids. Because FTIR deconvolutions (Fig. 2; Table 1) demonstrated that the enhanced α-helix in SP-Css ion-lock 1 or SP-Cff ion-lock 1 in lipids occurred concomitantly with reduced β-sheet, the charged ion-pair may partially augment α-helicity by disrupting β-sheet in the C-terminal region.

Interestingly, Fig. 3 shows that an analogous insertion of ion-locks in SP-C33 does not increase the α-helix levels of SP-C33 ion-lock 1 and SP-C33 ion-lock 2 in surfactant lipids, despite the relative PASTA energies for these daughter ion-locks being substantially higher than that of the parent SP-C33. The inability of ion-locks to further raise the

$\alpha$-helicity of SP-C33, unlike the large $\alpha$-helix increases seen for ion-lock incorporation into SP-Css and SP-Cff, may be due to the already high $\alpha$-helix and low $\beta$-sheet present in the SP-C33 peptide (Fig. 3). FTIR spectral deconvolutions for SP-C33, SP-C33 ion-lock 1 and SP-C33 ion-lock 2 in surfactant lipids each indicated $<\sim 8\%\beta$-sheet (Table 1), and additional recruitment of $\alpha$-helix from $\beta$-sheet residues may not be possible. In this context, SP-Css ion-lock 1 in surfactant lipids may be similarly optimized for high $\alpha$-helix, as FTIR deconvolutions showed elevated $\alpha$-helix and diminished $\beta$-sheet matching those of SP-C33 and SP-C33 ion-locks (Table 1).

## Prediction of transmembrane $\alpha$-helices for SP-C mimics using MPEx hydropathy calculations

Because secondary structure analysis using FTIR spectroscopy or PASTA do not directly measure membrane binding of helical peptides, the tendency of SP-C peptides to form transmembrane $\alpha$-helices was next evaluated with hydropathy calculations using the Membrane Protein Explorer (MPEx) program. MPEx determines the hydropathy (kcal/mol) for protein sequences using the augmented Wimley-White (WW) hydrophobicity scale and is based on measurements of the partitioning of hydrophobic pentapeptides and salt-bridge pairs into *n*-octanol (*Jayasinghe, Hristova & White, 2001*). Previous MPEx calculations have predicted transmembrane helices for protein sequences with high accuracy and are especially useful in preliminary characterizations of novel proteins or peptides (*Snider et al., 2009*; *Liang et al., 2012*). One important feature of MPEx is its option for engaging potential salt-bridges using an "enter-remove" icon on the control panel (*Snider et al., 2009*). However, note that dipalmitoylated SP-C is excluded from the below comparisons, because the MPEx and PASTA algorithms do not correct for how covalently attached, non-peptide moieties (e.g., the palmitoyl groups in native SP-C; Fig. 1) may modify membrane incorporation or peptide aggregation, or both.

As a surrogate for native SP-C, the rSP-C sequence was submitted to MPEx, which predicted that this SP-C mimic would insert into lipid bilayers as a transmembrane $\alpha$-helix with its most favorable hydropathy (i.e., 11.3 kcal/mol) through the C-terminal segment 12–30. The rSP-C sequence was chosen as a control for our MPEx analyses of SP-C ion-locks, because the recombinant protein is highly homologous with the dipalmitoylated SP-C (Fig. 1), lacks the two palmitoylcysteine groups of the native protein and has been previously subjected to structural and functional experiments. To simultaneously visualize the respective propensities of SP-C peptides to insert into lipids as transmembrane helix and to form $\beta$-sheet, the "Hydropathy from MPEx" (*ordinate*) is plotted *versus* the "Relative PASTA Energy" (*abscissa*) for various SP-C mimics (Fig. 4). The more positive hydropathy values (i.e., bottom-to-top) reflect greater incorporation as a transmembrane helix, while higher relative PASTA energy values (i.e., left-to-right) represent a lower tendency to form $\beta$-sheet. If rSP-C binds to lipid bilayers with the same helical conformation as in chloroform-methanol (i.e., $\alpha$-helix for residues 9–34) (*Luy et al., 2004*), then Fig. 4 predicts that rSP-C readily incorporates as a transmembrane $\alpha$-helix through its C-terminus. The very low PASTA energy indicated for rSP-C (Fig. 4) may also be responsible for the metastable helical properties observed in earlier 2D-NMR

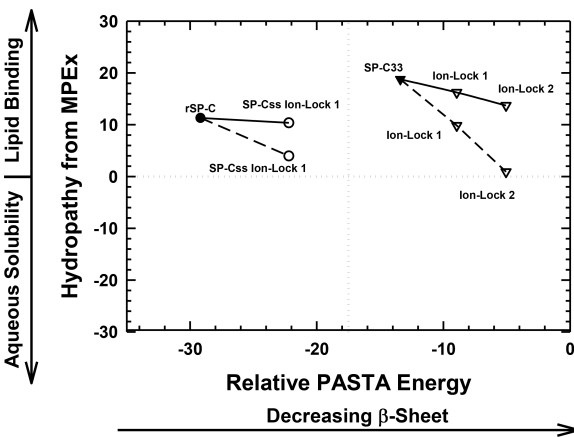

**Figure 4** **Plot of hydropathy (kcal/mol) vs. relative PASTA energy values, determined for the rSP-C, SP-Css ion-lock 1, SP-C33 and SP-C33 ion-lock 1 and SP-C33 ion-lock 2 peptides.** Hydropathy (kcal/mol) is a measure of the hydrophobic partitioning for helical peptides into transmembrane environments, determined using MPEx. Positive hydropathy predicts elevated lipid binding for helical peptides, while more negative values forecast greater water-solubility. Relative energy values were from PASTA, with less negative values reflecting a lower propensity to form $\beta$-sheet. *Top-left quadrant*: The rSP-C (*filled circle*) and SP-Css ion-lock 1 (*open circles*) values are shown. The PASTA energies for rSP-C and SP-Css ion-lock 1 are calculated from the respective segments 7–27, while MPEx hydropathies are determined from the respective segments 12–30. The SP-Css ion-lock 1 value, connected to the rSP-C value by a *solid line*, reflects the hydropathy for an $\alpha$-helical peptide with an engaged salt-bridge (i.e., close-range) between $Glu^-$–$Lys^+$. Alternatively, the SP-Css ion-lock 1 value, connected to the SP-C value by a *dashed line*, indicates a hydropathy with a disengaged salt-bridge (i.e., long-range). The SP-Cff ion-lock 1 values overlap those of SP-Css ion-lock 1. SP-Css and SP-Cff points are not included, because MPEx does not accurately predict the transmembrane insertion of such short helices (i.e., $\leq \sim 8$ residues) (Table 1). *Top-right quadrant*: The SP-C33 (*filled inverted triangles*) and SP-C33 ion-lock 1 and 2 (*open inverted triangles*) values are shown. The PASTA energies for SP-C33, SP-C33 ion-lock 1 and SP-C33 ion-lock 2 are each calculated from the C-terminal segment 6–26, while MPEx hydropathy is determined from each 13–31 segment. The SP-C33 ion-lock values, connected to the parent SP-C33 value by a *solid line*, reflect the hydropathies of $\alpha$-helical peptides with engaged salt-bridges between the $Glu^-$–$Lys^+$ pair. Alternatively, the SP-C33 ion-lock values, connected to the SP-C33 value by *dashed lines*, indicate hydropathies with disengaged salt-bridges between the $Glu^-$–$Lys^+$ pairs. See Fig. 1 for sequences.

studies (*Luy et al., 2004*), as $\beta$-sheet multimers or fibrils should here be thermodynamically favored over $\alpha$-helix (Fig. 3) (*Trovato et al., 2006*; *Trovato, Seno & Tosatto, 2007*).

The relative tendencies of SP-Css ion-lock 1 and SP-Cff ion-lock 1 to penetrate into lipid bilayers as transmembrane $\alpha$-helices were also predicted with the joint MPEx and PASTA plot in Fig. 4. Earlier MPEx research showed that hydrophobic helices bearing salt-bridges may penetrate deeply into lipid bilayers, because there is only a slight thermodynamic penalty for replacing the hydrophobic residues of a transmembrane sequence with a neutral, charged ion-pair (e.g., $Glu^-$ and $Lys^+$) (*Snider et al., 2009*; *Jayasinghe, Hristova & White, 2001*). Moreover, previous electrostatic calculations indicated that salt-bridges with close-range distances ($\leq 4.0$ Å) between the cationic residue "N" and the nearest anionic residue "O" are mostly $\alpha$-helix stabilizing, while disengaged salt-bridges with long-range distances ($> 5$ Å) may be helix destabilizing (*Kumar & Nussinov, 1999*; *Kumar & Nussinov, 2002*). As indicated above, MPEx allows hydropathic analysis of SP-C ion-lock sequences in membranes assuming the salt-bridge is either engaged (i.e., close-range salt-bridge)

or disengaged (i.e., long-range salt-bridge). With a close-range salt-bridge between the Glu$^-$-20–Lys$^+$-24 pair, MPEx forecasts that a highly $\alpha$-helical SP-Css ion-lock 1 would deeply embed into the lipid bilayer due to the high hydropathy (i.e., $\sim$10.3 kcal/mol) of its segment 12–30. Alternatively, with a long-range salt-bridge between Glu$^-$-20–Lys$^+$-24, the corresponding MPEx hydropathy would be much lower (i.e., $\sim$4.0 kcal/mol) and predicts a more superficial association between a less helical SP-Css ion-lock 1 and the bilayer. Our FTIR results indicating high $\alpha$-helicity for SP-Css ion-lock 1 in surfactant lipids are consistent with a close-range salt-bridge forming in this environment, permitting deep insertion of this SP-C mimic as a transmembrane $\alpha$-helix. Specifically, FTIR deconvolution in Table 1 showed that an SP-Css ion-lock 1 in surfactant lipids is highly $\alpha$-helical ($\sim$63.1%). Thus, the SP-Css ion-lock 1 $\alpha$-helix ($\sim$21 residues) may span the width of the bilayer, as it lies within the 17–25 residue window reported for other transmembrane $\alpha$-helices (*Liang et al., 2012*). The PASTA prediction that the Glu$^-$-20–Lys$^+$-24 pair in SP-Css ion-lock 1 converts segment 7–27 from $\beta$-sheet to $\alpha$-helix (Figs. 3 and 4) also supports the MPEx assignment that $\alpha$-helix is confined to the C-terminal region. Consequently, helical SP-Css ion-lock 1 with a close-range salt-bridge shows a high hydropathy, which is only slightly less than that of rSP-C (i.e., 10.3 vs. 11.3 kcal/mol in Fig. 4), and should mimic the transmembrane incorporation of native SP-C peptides. On the other hand, the reduced $\alpha$-helicity observed for SP-Cff ion-lock 1 in surfactant lipids is compatible with a disruptive, long-range salt-bridge that lowers $\alpha$-helical levels to $\sim$44.6% (Table 1). The resulting SP-Cff ion-lock 1 $\alpha$-helix is too short ($\sim$15 residues) to span the width of the bilayer (*Liang et al., 2012*) and instead may be limited to a single monolayer. These studies suggest that the replacement of the two serines in SP-Css ion-lock 1 with vicinal phenylalanines disorders the close-range, salt-bridge between Glu$^-$-20–Lys$^+$-24, through either direct or indirect mechanisms.

FTIR secondary structure and theoretical analyses similarly examined the insertion of SP-C33 mimics into the lipid bilayer as transmembrane $\alpha$-helices. Figure 4 indicates higher MPEx and PASTA energy values for SP-C33 relative to those of rSP-C, predicting that the swap of poly-Leu with the poly-Val sequence of native SP-C produces an SP-C33 mimic that associates with lipids as a transmembrane $\alpha$-helix. In support of this assignment, FTIR showed that the principal spectral component for SP-C33 UCLA in surfactant lipids was $\alpha$-helix and that the $\alpha$-helicity (67.2% from Table 1) for SP-C33 UCLA is sufficiently high for this SP-C mimic to incorporate as a transmembrane helix ($\sim$22 residues long). In this context, Fig. 4 also predicts that SP-C33 ion-lock 1 or SP-C33 ion-lock 2 (i.e., SP-C33 mimics with one or two Glu$^-$-20–Lys$^+$-24 pairs; Fig. 1) will incorporate into lipid bilayers either as transmembrane $\alpha$-helices with close-range salt-bridges or disordered $\alpha$-helices with long-range salt-bridges and reduced membrane affinity. Because our FTIR results show only high $\alpha$-helix levels for these SP-C33 ion-locks matching those of the parent SP-C33 in surfactant lipids (Table 1), both SP-C33 ion-lock 1 and SP-C33 ion-lock 2 will likely form engaged salt-bridges, and incorporate into surfactant lipids as transmembrane $\alpha$-helices.

## Orientation-dependent FTIR analysis of the helical axes of SP-C mimics in surfactant lipids

The topography of helical SP-C mimics in surfactant lipid multilayers was experimentally assessed by recording polarized infrared spectra of each lipid-peptide film that was rotated from 0° to 90°. Orientation-dependent FTIR spectroscopy of native SP-C (*Vandenbussche et al., 1992*; *Pastrana, Maulone & Mendelsohn, 1991*) and unrelated membrane proteins (*Chia et al., 2002*; *Beevers & Kukol, 2006*) was previously used to determine the Θ angles between their respective helical axes and the normal to the lipid bilayer plane. Early polarized FTIR studies indicated that native bovine and porcine SP-C proteins (*Vandenbussche et al., 1992*; *Pastrana, Maulone & Mendelsohn, 1991*) each insert into surfactant lipids as a transmembrane helix whose long molecular axis parallels the phospholipid acyl chains and a maximum angle of ∼20–24° with respect to the bilayer normal. In the present research, the polarized FTIR spectra of the amide I band for SP-Css ion-lock 1 in surfactant multilayers were highly anisotropic, with high and low absorbance at 0° and 90° (Fig. 5A). The dichroic ratio R for SP-Css ion-lock 1 is computed as the ratio of the integrated amide I absorption at parallel and perpendicular with polarized infrared light. The use of this dichroic R indicated a maximum tilt angle $\Theta = 22.6°$ for the helical axis of SP-Css ion-lock 1 with respect to the bilayer normal (Table 1), similar to those determined previously for native SP-C (*Vandenbussche et al., 1992*; *Pastrana, Maulone & Mendelsohn, 1991*). Based on its high $\alpha$-helicity (63.1%) (Fig. 2A; Table 1), the SP-Css ion-lock 1 probably incorporates in surfactant lipids as a transmembrane $\alpha$-helix (∼21 residues long) spanning the bilayer with a maximum 22.6° tilt, particularly if this SP-C mimic is highly hydropathic due to a salt-bridge between Glu$^-$-20–Lys$^+$-24 (Fig. 4). When compared with FTIR results from SP-Css ion-lock 1 or native SP-C, however, the polarized FTIR spectra for SP-Cff ion-lock 1 in surfactant lipids were less anisotropic with a lower dichroic R and demonstrated a correspondingly higher maximum tilt angle ($\Theta = 34.2°$) (Table 1). Because of its reduced $\alpha$-helix levels (∼44.6%) calculated from FTIR spectral deconvolutions and higher maximum tilt angle (Figs. 2A and 5B; Table 1), the relatively short (∼15 residues) $\alpha$-helix of SP-Cff ion-lock 1 in surfactant lipids is probably limited to one bilayer leaflet. One hypothesis worth exploring is that the low $\alpha$-helicity and more oblique Θ observed for SP-Cff ion-lock is due to a failure of its Glu$^-$-20–Lys$^+$-24 to form an engaged salt-bridge (i.e., close-range) in surfactant lipids.

Analogous oriented FTIR experiments were performed to characterize the membrane topography of SP-C33 and SP-C33 ion-locks in surfactant lipids. As noted above, conventional FTIR analysis of SP-C33 UCLA, SP-C33 ion-lock 1 or SP-C33 ion-lock 2 (Fig. 1) in surfactant lipids each showed spectra with a predominant $\alpha$-helix band centered at 1655 cm$^{-1}$ (Fig. 2B) and elevated levels (65%–67%) of $\alpha$-helix (Table 1). The polarized FTIR spectra of the amide I bands for SP-C33 UCLA (Fig. 5D), SP-C33 ion-lock 1 (Fig. 5C) and SP-C33 ion-lock 2 (*not shown*) were all highly anisotropic, indicating maximum Θ angles of ∼23° (Table 1). Consequently, SP-C33 UCLA and the SP-C33 ion-locks incorporate into surfactant lipid as transmembrane $\alpha$-helices (∼21–22 residues long) spanning the bilayer with a maximum tilt angle of ∼23°, confirming that they

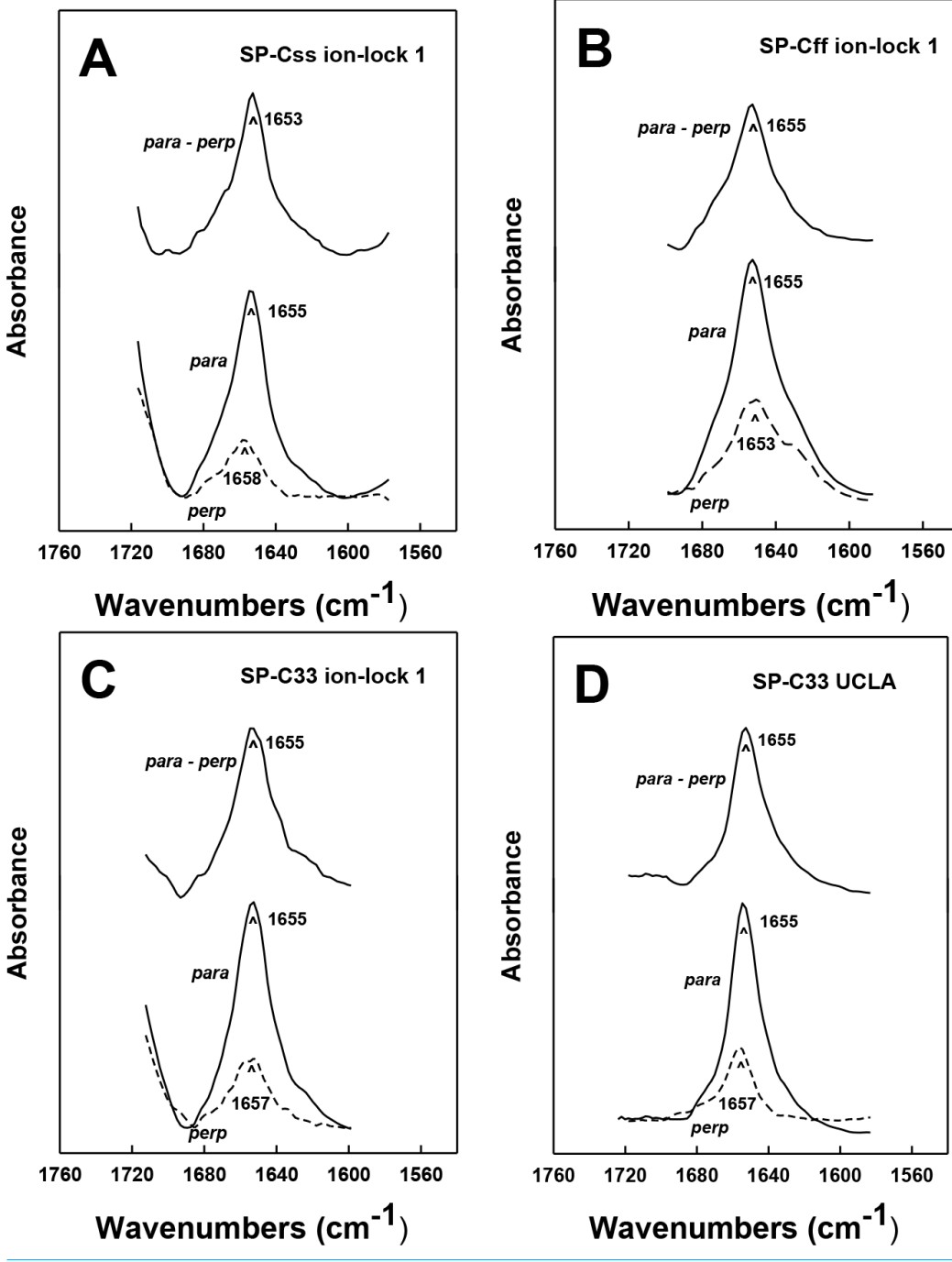

**Figure 5** ATR-FTIR polarized spectra of the amide I band for synthetic SP-C peptide mimics in oriented DPPC:POPC:POPG (5:3:2) bilayers with 10 mM deuterated phosphate buffer (pD 7.4). The bottom spectra in each plate were recorded with parallel (*para*; solid line) and perpendicular (*perp*; dashed line) polarized light with respect to the incident plane. The dichroic spectra at top in each plate is the difference between the spectrum recorded with parallel polarized light from that obtained with perpendicular polarized light. Recorded and subtracted spectra for each SP-C peptide are drawn using the same ordinate scale. (A) SP-Css ion-lock 1. (B) SP-Cff ion-lock 1. (C) SP-C33 ion-lock 1. (D) SP-C33 UCLA. See Fig. 1 for SP-C mimic sequences.

accurately reproduce the membrane topography of native SP-C proteins (*Vandenbussche et al., 1992*; *Pastrana, Maulone & Mendelsohn, 1991*). Moreover, SP-C33 ion-lock 1 and SP-C33 ion-lock 2 each probably contain stable salt-bridges as predicted from MPEx-hydropathy analysis (Fig. 4; *top right quadrant*), which would facilitate the deep penetration of these SP-C mimics into surfactant bilayers. Importantly, our orientation-dependent FTIR results with SP-C33 UCLA (Fig. 5D; Table 1) are in good agreement with those earlier reported for SP-C33 incorporated into surfactant lipid bilayers (*Almlen et al., 2011*), despite these studies using different experimental conditions for peptide synthesis and FTIR spectral measurements.

## FTIR analysis of hydrogen/deuterium (H/D) exchange for amide protons of SP-C mimics in surfactant lipids

For selected SP-C mimics with surfactant lipids, the extent of peptide helix insertion into the lipid bilayer was characterized by measuring the hydrogen/deuterium (H/D) exchange of amide protons. Previous studies of native SP-C (*Vandenbussche et al., 1992*; *Pastrana, Maulone & Mendelsohn, 1991*) and unrelated membrane proteins (*Chia et al., 2002*; *Grimard et al., 2001*) indicated that protons participating in the H-bonds of highly ordered structures such as the transmembrane $\alpha$-helix may show a lack of deuterium exchange for up to 6–28 h. The H/D exchange for amide protons that maintain the transmembrane $\alpha$-helix was assessed as follows. SP-C mimic-lipid samples on the ATR crystal were hydrated by bubbling nitrogen through a 10 mM deuterated phosphate buffer (pD 7.4) at 37 °C, and the amide I and amide II peaks were recorded with FTIR spectra over the range 1500–1700 cm$^{-1}$ as a function of time (0–6 h). The peak between 1600 and 1700 cm$^{-1}$ reflects the amide I band (i.e., $\nu$ (C=O) of the peptide bond) that is sensitive to the secondary conformations of the protein, while the second band between 1525 and 1565 cm$^{-1}$ is primarily due to amide II (i.e., $\delta$ (N–H) of the peptide bond) (*Vandenbussche et al., 1992*; *Jackson & Mantsch, 1995*). The ratio of the amide II to amide I peak areas is proportional to the H/D exchange, as such exchange would shift the amide II band (at $\sim$1543 cm$^{-1}$) to that of amide II' (at $\sim$1450 cm$^{-1}$) (*Pastrana, Maulone & Mendelsohn, 1991*). Contrarily, the absence of H/D exchange would suggest that the H-bonds are isolated from aqueous media (e.g., transmembrane $\alpha$-helices), or are in aqueous environments in ordered conformations where exchange is slow.

In Fig. 6 the percentage (%) unexchanged hydrogens for the SP-Css ion-lock 1 in surfactant lipids is shown as a function of time (0–6). Analogous H/D curves for membrane proteins were previously analyzed as the sum of multiple linear exponential (first-order) decays (*Grimard et al., 2001*). Besides fast- and intermediate-H/D exchange compartments (i.e., with $t_{1/2}$'s between 10 and 120 min involving $\sim$49% of the helical residues), there is a final compartment for helical amide hydrogens ($\sim$51%) that resists H/D exchange for $t \leq 6$ h (Fig. 6). The inaccessible fraction at long-times is probably due to segments buried deeply as transmembrane $\alpha$-helix, as the residual amide II band remains centered at $\sim$1543 cm$^{-1}$ (*not shown*). Because spectral deconvolutions indicated $\sim$21 $\alpha$-helical residues for SP-Css ion-lock 1 in surfactant lipids (Table 1), $\sim$11 $\alpha$-helical

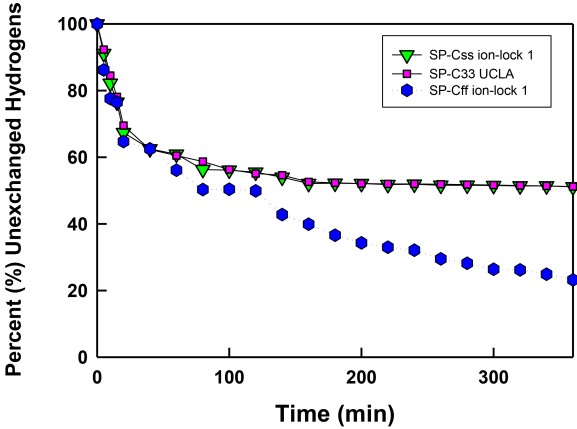

**Figure 6 The percent (%) unexchanged hydrogens vs. time (min) from FTIR of SP-C mimics in surfactant lipids.** SP-Css ion-lock 1 (*inverted triangle, green*), SP-C33 UCLA (*square, magenta*) and SP-Cff ion-lock 1 (*hex, blue*) were each incorporated into DPPC:POPC:POPG (5:3:2, weight ratio), and the peptide-lipid sample on the ATR crystal was initially hydrated by bubbling nitrogen through a 10 mM deuterated phosphate buffer (pD 7.4) at 37 °C. The time-dependent H/D exchange of the amide group was determined from the decay of the amide II band area (1525–1565 cm$^{-1}$) with respect to the corresponding amide I band area (~1600–1700 cm$^{-1}$) as a function of time (0–6 h).

residues are readily calculated as unexchanged for this SP-C mimic at long-times (*Almlen et al., 2011*; *Beevers & Kukol, 2006*). Importantly, similar H/D exchange kinetics was earlier reported for bovine and porcine SP-C (*Vandenbussche et al., 1992*; *Pastrana, Maulone & Mendelsohn, 1991*), indicating that SP-Css ion-lock 1 is accurately mimicking the ability of native SP-C proteins to embed into surfactant lipids as transmembrane α-helices. In additional studies, the kinetic H/D exchange profile for SP-Css ion-lock 1 was nearly identical to that of SP-C33 UCLA (Fig. 6), and very similar to that reported earlier for the conventional SP-C33 (*Almlen et al., 2011*). The comparable H/D exchange and maximum tilt angles observed for SP-Css ion-lock 1, SP-C33 UCLA and SP-C33 (Figs. 5 and 6; Table 1) (*Almlen et al., 2011*) indicate that these three SP-C mimics similarly incorporate into surfactant lipids as transmembrane α-helices. Interestingly, the H/D curve for SP-Cff ion-lock 1 in Fig. 6 demonstrates both similarities and differences with those of SP-Css ion-lock 1 and SP-C33 UCLA. Despite SP-Cff ion-lock 1 showing fast-, intermediate- and very slow-exchange compartments analogous to those of the other two SP-C mimics, there was overall a more rapid H/D exchange for SP-Cff ion-lock 1 in Fig. 6. These discrepancies were particularly noticeable at 6 h, when the unexchanged hydrogen component for SP-Cff ion-lock 1 was ~23.2%, while those for SP-Css ion-lock 1 and SP-C33 UCLA were each ~51% (Fig. 6). Consequently, only ~3.5 helical residues of SP-Cff ion-lock 1 showed unexchanged hydrogen at long-times in Fig. 6, compared to the ~11 helical residues that remain unexchanged in either SP-Css ion-lock 1 or SP-C33 UCLA. The sharply lower unexchanged helical protons for SP-Cff ion-lock 1 in surfactant lipids (Fig. 6) may be due to a more flexible membrane α-helical component or increased accessibility of helical amide hydrogens to the aqueous medium.

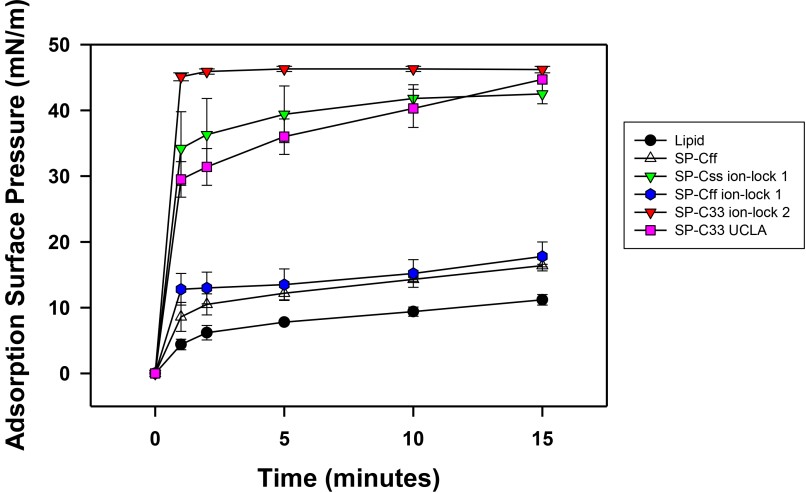

**Figure 7 Adsorption of synthetic surfactant compositions containing SP-Cff, SP-C33 and SP-C ion-lock mimics plus synthetic phospholipids (5:3:2 weight ratio DPPC:POPC:POPG).** Adsorption was measured as Adsorption Surface Pressure (mN/m), following injection of a surfactant bolus beneath the interface of a stirred subphase at time zero. Surface pressure is the amount of surface tension lowering below that of the pure subphase (normal saline adjusted to pH 7.0) at 37 ± 0.5 °C. Higher surface pressure equates to lower surface tension. Surfactant concentration was 0.0625 mg phospholipid/ml of subphase. Data are Means ± SEM for $n = 4$. Sequences for SP-Cff, SP-Css ion-lock 1, SP-Cff ion-lock 1, SP-C33 ion-lock 2 and SP-C33 UCLA are given in Fig. 1.

## *In vitro* surface activity determined with adsorption surface pressure measurements

Synthetic surfactant preparations were formulated by mixing synthetic lipids (Lipids) consisting of 5:3:2 (weight ratio) DPPC:POPC:POPG, with 3.0% by weight SP-Css ion-lock 1, SP-Cff ion-lock 1, SP-Cff, SP-C33 ion-lock 1, SP-C33 UCLA (positive control) and Lipids (negative control). Here, adsorption was determined as the Adsorption Surface Pressure (mN/m) as a function of time (0–15 min), with higher surface pressures reflecting lower surface tensions. The time-course plots in Fig. 7 indicate that each of these SP-C mimics attained maximal adsorption surface pressure within ∼5–10 min. The relative order for the surfactant activities of the various preparations was as follows: SP-C33 ion-lock 2 ≥ SP-Css ion-lock 1, SP-C33 UCLA ≫ SP-Cff ion-lock 1 ∼ SP-Cff > Lipids. Previous *in vitro* studies on SP-C(Leu), a homologous precursor to our SP-C33 UCLA, with a lipid mixture of DPPC/POPG (7:3 ratio by mass) showed similarly enhanced adsorption surface pressure curves (*Nilsson et al., 1998*).

## *In vitro* dynamic surface activity determined with captive bubble surfactometry

Synthetic surfactant formulations were also prepared for *in vitro* experiments with captive bubble surfactometry and *in vivo* studies using ventilated, lung-lavage rabbits by mixing synthetic lipids (Lipids), consisting of 5:3:2 (weight ratio) DPPC:POPC:POPG, with 3.0% by weight SP-Css ion-lock 1, SP-Cff ion-lock 1, SP-C33 ion-lock 1, SP-Cff, SP-Css, SP-C33 UCLA (positive control) and Lipids (negative control). The SP-Css ion-lock 1 surfactant

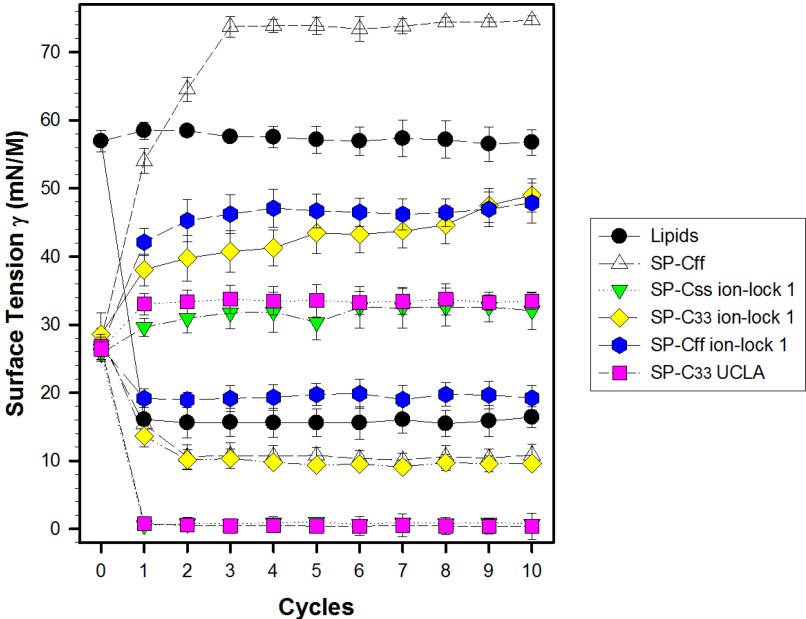

**Figure 8 Surface activity of synthetic surfactants on the captive bubble surfactometer.** Minimum and maximum surface tension values are plotted for synthetic lipids (Lipids) with 3.0% by weight SP-Cff, SP-Css, SP-Css ion-lock 1, SP-C33 ion-lock 1, SP-Cff ion-lock 1 and SP-C33 UCLA (positive control) or synthetic lipids alone (negative control) for 10 successive compression cycles on a captive bubble surfactometer (20 cycles/min, 37 °C). Synthetic lipids = 5:3:2 (weight ratio) DPPC:POPC:POPG. Values shown are means with estimated SEM, for $n = 4$–6. See Fig. 1 for SP-C mimic sequences.

had very high surface activity in captive bubble experiments and reached minimum surface tension values <1 mN/m during each of ten consecutive cycles of dynamic cycling (rate of 20 cycles/min, Fig. 8). Importantly, our SP-C33 UCLA (positive control) attained low surface tension values <1 mN/m that not only were identical to those of SP-Css ion-lock 1 (Fig. 8), but also matched those of SP-C33 in earlier captive bubble experiments with DPPC/POPG lipid mixtures. The SP-C33 ion-lock 1 surfactant exerted similar low surface tensions as SP-C33 UCLA. Contrarily, SP-Cff ion-lock 1, Lipids (negative control), SP-Cff and SP-Css all reached significantly higher minimum surface tension values of 19, 16, 11 and 14 mN/m ($p < 0.001$) versus SP-Css ion-lock 1 and SP-C33 UCLA after ten cycles on the captive bubble surfactometer (Fig. 8). In summary, the relative order for the surfactant activities determined with captive bubble surfactometry for the various preparations was as follows: SP-Css ion-lock 1 ∼ SP-C33 UCLA ∼ SP-C33 ion-lock 1 ≫ SP-Cff ∼ SP-Css ≫ Lipids ∼ SP-Cff ion-lock 1.

### *In vivo* activity of synthetic surfactants in ventilated, lung-lavaged, surfactant-deficient rabbits

The pulmonary activity of SP-C mimic surfactants (described above for captive bubble experiments) was investigated in comparison to SP-C33 UCLA (positive control) and Lipids alone (negative control) during a 120 min period following intratracheal instillation of these surfactants into ventilated rabbits with ARDS induced by *in vivo*

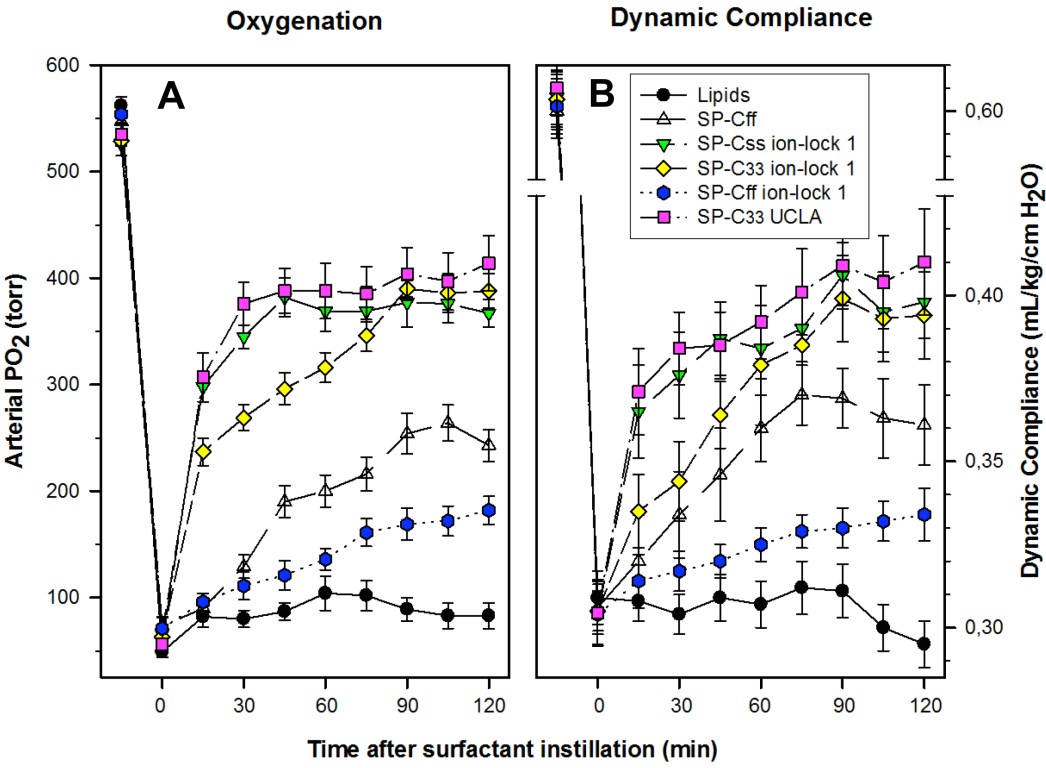

**Figure 9 Arterial oxygenation and dynamic compliance in surfactant-treated, ventilated rabbits with ARDS induced by *in vivo* lavage.** (A) Arterial partial pressure of oxygen ($PaO_2$ in torr) shown as a function of time for groups of at least 4 rabbits treated with synthetic lung surfactants containing synthetic lipids with 3.0% by weight SP-Cff, SP-Css ion-lock 1, SP-C33 ion-lock 1, SP-Cff ion-lock 1, and SP-C33 UCLA (positive control), or synthetic lipids alone (negative control). Synthetic lipids are DPPC:POPC:POPG (5:3:2, weight ratio). (B) Dynamic compliance (mL/kg/cm $H_2O$) shown as a function of time for groups of at least 4 rabbits treated with the same lung surfactant preparations as shown in (A). Data are shown as means $\pm$ SEM, for $n = 4$–7. Sequences for SP-C mimics are shown in Fig. 1.

lavage (Fig. 9). Oxygenation and lung compliance increased quickly to a plateau phase (within ∼30–60 min) after instillation of SP-Css ion-lock 1 and SP-C33 UCLA, with SP-C33 ion-lock 1 reaching similarly high functional plateaus after somewhat longer times (within ∼60 min) (Fig. 9). On the other hand, the SP-C mimics SP-Cff and SP-Cff ion-lock 1 restored more slowly (within ∼90 min) and only partially the oxygenation and dynamic compliance observed with SP-Css ion-lock 1 and SP-C33 UCLA. The corresponding oxygenation and dynamic compliance curves obtained for SP-Css (not shown for clarity) overlapped those of SP-Cff in Fig. 9. Last, instillation of the negative control of Lipids alone had minimal effects on arterial oxygenation or compliance. The relative order of pulmonary activity in terms of both oxygenation and compliance ($\geq$90 min) was given as: SP-Css ion-lock 1 ∼ SP-C33 UCLA (positive control) ∼ SP-C33 ion-lock 1 ≫ SP-Cff ∼ SP-Css > SP-Cff ion-lock 1 ≫ Lipids (negative control). Although the differences in oxygenation and compliance between SP-Css ion-lock 1, SP-C33 UCLA and SP-C33 ion-lock 1 were not statistically significant starting at 90 min after surfactant instillation, each of these SP-C mimics consistently exceeded the corresponding

performance of SP-Cff, SP-Css, SP-Cff ion-lock 1 and lipids only surfactants ($p < 0.01$) (Fig. 9).

## DISCUSSION

Structural and functional studies investigated *first generation* SP-C peptides based on native SP-C sequences (e.g., synthetic SP-Cff and SP-Css lacking palmitoyl groups; Fig. 1), and assessed their utility as scaffolds for new SP-C ion-lock peptides with enhanced helicity and surfactant activities. Unlike the high $\alpha$-helix observed for native dipalmitoylated SP-C in surfactant lipids (*Vandenbussche et al., 1992*; *Pastrana, Maulone & Mendelsohn, 1991*), FTIR spectroscopy for either SP-Css or SP-Cff in lipids with aqueous buffer showed primarily $\beta$-sheet (Fig. 2A; Table 1). Comparable $\beta$-structures also predominated for SP-Css and S-Cff in either 100% aqueous buffer or 65% HFIP/35% aqueous buffer (Table 2), indicating that these SP-C mimic conformations are insensitive to the polarity of their environment. These FTIR spectral results are probably due to the strong parallel $\beta$-sheet propensities predicted by the PASTA analysis for SP-Css or SP-Cff in Fig. 3. The above findings also suggest that palmitoylcysteines in native SP-C at least partially act as helical 'adjuvants' to override the $\beta$-sheet propensities of its primary sequence (*Walther et al., 2010*; *Conchillo-Sole et al., 2007*). With this background, it was surprising that aqueous HFIP did not substantially raise the $\alpha$-helicity of SP-Css or SP-Cff over that in aqueous buffer (Table 2), especially because HFIP is a strong helix-inducing and stabilizing cosolvent (*Hirota, Mizuno & Goto, 1997*) and provides an apolar environment that simulates lipid membranes (*Crescenzi et al., 2002*). HFIP protects against $\beta$-sheet formation by "Teflon® Coating" exposed $\alpha$-helices (*Rajan et al., 1997*; *Crescenzi et al., 2002*; *Roccatano et al., 2005*), thereby preventing the water attack on the helical H-bonds that are required for conversion to $\beta$-sheet. Consequently, the elevated $\beta$-sheet observed for SP-Css and SP-Cff in aqueous HFIP (Table 2) may be due to the extraordinarily high $\beta$-sheet propensities of these SP-C mimics (Fig. 3) outweighing the helix-promoting actions of HFIP.

Comparisons with previous FTIR results (*Zandomeneghi et al., 2004*; *Cerf et al., 2009*) additionally suggest that SP-Css and SP-Cff form 'amyloid-like', $\beta$-sheet fibrils in surfactant lipids, and that these fibrils may be responsible for the low surfactant activities observed with SP-C mimics (Figs. 7–9). An earlier literature review indicated that native $\beta$-sheet proteins typically produce FTIR spectra with maxima at frequencies above $\sim$1630 cm$^{-1}$, while the corresponding spectra of amyloid fibrils demonstrate maxima at $\sim$1630 cm$^{-1}$ and lower frequencies (*Zandomeneghi et al., 2004*). Following this scheme, the FTIR spectra in Fig. 2A suggest that SP-Css and SP-Cff with lipids may each be forming amyloidogenic fibrils. Further comparisons with FTIR analyses of the amyloid $A\beta(1–42)$ suggested that SP-Css or SP-Cff in lipids are producing fibrils (*Cerf et al., 2009*). Under conditions that promote fibrils, FTIR spectra of $A\beta(1–42)$ exhibited a single peak at $\sim$1630 cm$^{-1}$ and negligible absorption at $\sim$1695 cm$^{-1}$, and fibrils were assigned to $A\beta(1–42)$ forming parallel $\beta$-sheet. Contrarily, under conditions favoring smaller $A\beta(1–42)$ oligomers, FTIR spectra showed a primary peak at $\sim$1630 cm$^{-1}$ and
a significant minor peak (i.e., ∼1/5th the intensity of the major component) that were attributed to $A\beta(1–42)$ oligomers folding like antiparallel $\beta$-sheet (*Cerf et al., 2009*). Because the FTIR spectra of SP-Css and SP-Cff in surfactant lipid (Fig. 2A) are most comparable to those of fibrillar $A\beta(1–42)$ (*Cerf et al., 2009*), our SP-C mimics in surfactant lipids may similarly adopt parallel $\beta$-sheet, fibril structures. This assignment agrees with PASTA predictions that the C-terminal regions of SP-Css and SP-Cff have high parallel $\beta$-sheet propensities (Figs. 3 and 4). Moreover, prior experiments indicate that the homologous rSP-C (Fig. 1) readily transforms from $\alpha$-helical conformers to $\beta$-sheet fibrils in lipid-mimics (*Luy et al., 2004*). Any $\beta$-sheet fibrils occurring in lipid mixtures of SP-Css or SP-Cff would not necessarily be toxic but instead may represent an inert peptide reservoir. Recent evidence with the amyloid $A\beta$ protein demonstrates that it is only small molecular weight oligomers, but not monomers or high molecular weight fibrils, that are cytotoxic (*Haass & Selkoe, 2007*; *Nerelius, Fitzen & Johansson, 2010*). In this context, SP-Cff and/or SP-Css in lipids showed surfactant activities that were between those of lipids alone (negative control) and those with the more active SP-C33 UCLA (positive control) (see below and Figs. 7–9). The relatively low surfactant activities for SP-Cff and SP-Css with *in vitro* (Figs. 7 and 8) and *in vivo* (Fig. 9) assays may be due to these SP-C mimics forming inactive fibrils consisting of parallel $\beta$-sheet, thus effectively lowering the concentrations of surface-active SP-C peptides in helical conformations.

Because the above *first generation* SP-C peptides were unsuccessful in reproducing key structural and functional properties of the native protein in surfactant lipids, a *second generation* of SP-C mimics without palmitoyls is now being developed that seeks to increase $\alpha$-helicity and surfactant activities through amino acid replacements. Previous experimental and theoretical studies indicated that SP-C embeds deeply into lipid bilayers, in which its helical axis and covalently attached palmitoyls are nearly parallel to the fatty acyl chains, while its cationic N-terminal region is restricted to the polar headgroup region (see Introduction). As enhanced *in vitro* and *in vivo* surfactant activities are highly correlated with elevated $\alpha$-helix content, a biomimetic approach was used to design $\alpha$-helicity and other essential molecular features of native SP-C into new SP-C analogs. One such class of SP-C mimics is the SP-C33 peptide, produced with the porcine SP-C sequence as a template, in which ten valines were replaced by leucines, the two palmitoylcysteines substituted with serines and the N-terminal arginine was moved closer to the poly-Leu sequence (Fig. 1) (*Johansson et al., 2003*). This synthetic SP-C mimic showed *in vitro* and *in vivo* surfactant activities equivalent to those of native SP-C (*Johansson et al., 2003*; *Almlen et al., 2011*), which may be due to both SP-C33 and native SP-C similarly inserting into surfactant lipids with slightly tilted transmembrane $\alpha$-helices (*Pastrana, Maulone & Mendelsohn, 1991*; *Vandenbussche et al., 1992*; *Almlen et al., 2011*).

We performed experiments here to determine whether an SP-C33 peptide with high surfactant activity was appropriate as a positive control to benchmark the structural and surfactant properties of other *second generation* SP-C peptides. With respect to the SP-C33 class, studies were conducted with 'SP-C33 UCLA', an SP-C mimic with the same sequence as SP-C33 (Fig. 1) but synthesized using a separate protocol. For example, FTIR spectra of

SP-C33 UCLA in aqueous buffer, HFIP-aqueous buffer and surfactant lipids in aqueous buffer indicated that the predominant conformation was $\alpha$-helix (Fig. 2B; Tables 1 and 2). These spectral results agree with prior findings showing high $\alpha$-helix for SP-C33 in surfactant lipids and lipid-mimics (i.e., methanol or detergent micelles) using CD or FTIR spectroscopy (*Johansson et al., 2003*; *Almlen et al., 2011*). The enhanced $\alpha$-helix levels for SP-C33 UCLA and SP-C33 are consistent with PASTA predictions that SP-C33 sequences (Fig. 1) have a lower propensity to form $\beta$-sheet than native SP-C peptides (Figs. 3 and 4). The total replacement of the poly-valine sequence with poly-leucine in SP-C33 UCLA suppresses 'amyloid-like' $\beta$-sheet structures by favoring $\alpha$-helix and eliminates the unstable conformations associated with SP-C lacking palmitoyl groups. Orientation FTIR and H/D exchange experiments indicated that SP-C33 UCLA incorporates into lipid bilayers as a slightly tilted transmembrane $\alpha$-helix (Figs. 5 and 6; Table 1), demonstrating that SP-C33 UCLA accurately mimics the membrane topography of both SP-C33 (*Almlen et al., 2011*) and native dipalmitoylated SP-C (*Pastrana, Maulone & Mendelsohn, 1991*; *Vandenbussche et al., 1992*). Also supporting the above experimental results are MPEx hydropathy predictions that SP-C33 sequences insert into membrane bilayers when folded as $\alpha$-helices (Fig. 4). In this context, note that SP-C33 UCLA demonstrated high *in vitro* (Figs. 7 and 8) and *in vivo* (Fig. 9) surfactant activities, consistent with the elevated surfactant activities reported earlier for SP-C33 in comparisons with commercial preparations containing native SP-C (*Johansson et al., 2003*; *Almlen et al., 2011*). The tilted transmembrane $\alpha$-helix, identified here for SP-C33 UCLA in lipid bilayers (Figs. 5 and 6; Table 1) and earlier for SP-C33 (*Almlen et al., 2011*), likely represents the surface-active arrangement shared by native SP-C, and departures from this membrane topography may reduce surfactant properties.

The principal hypothesis to be tested in this paper was that the selective insertion of salt-bridges may produce SP-C mimics that insert into lipid bilayers as slightly tilted, transmembrane $\alpha$-helices with high surfactant activity. In partial support of this hypothesis, prior reports indicate that engaged salt-bridges (or ion-locks) in either designer peptides or native membrane proteins promote $\alpha$-helix. With CD studies of aqueous synthetic peptides, for example, incorporation of Glu$^-$ and Lys$^+$, Asp$^-$ and Lys$^+$, Asp$^-$ and Arg$^+$ or Glu$^-$ and Arg$^+$ spaced at $(i + 4)$ intervals in the sequence was helix stabilizing (*Marqusee & Baldwin, 1987*). This enhanced $\alpha$-helicity for aqueous peptides is at least partly due to the formation of an electrostatically neutral ion-pair *via* the positive- and negative-side groups for the various amino-acid pairings (*Kumar & Nussinov, 1999*; *Kumar & Nussinov, 2002*). Employing an augmented Wimley-White (WW) hydrophobicity scale, MPEx analyses predicted that 'ion-locks' such as Glu$^-$–Lys$^+$ increase the helicity and lipid partitioning of peptides into membranes (*Snider et al., 2009*). Additionally, continuum electrostatic calculations indicated that salt-bridges with close distances ($\leq 4.0$ Å) between the cationic residue "N" and the nearest anionic residue "O" are mostly $\alpha$-helix stabilizing, while those with correspondingly further distances ($> 5.0$ Å) may be helix destabilizing (*Kumar & Nussinov, 1999*; *Kumar & Nussinov, 2002*). Earlier surveys of PDB-deposited structures have reported salt-bridges with a close-range

cutoff ($\leq$4.0 Å) in numerous proteins (*Kumar & Nussinov, 1999*; *Kumar & Nussinov, 2002*) including $Ca^{2+}$-ATPase (*Toyoshima et al., 2000*) (PDB: 1SU4) with a membrane helix stabilized by an intrahelical charge-pair (e.g., $Asp^{-}$-59–$Arg^{+}$-63) when facing lipids (*Bano-Polo et al., 2012*). Importantly, ion-locks have been used in several designer peptides as a 'Molecular Velcro®' for specific purposes, such as enhancement of $\alpha$-helix and anti-HIV activity in sifuvirtide (*He et al., 2008*), and the trapping of otherwise unstable structural features such as $\pi$-helix (*Chapman et al., 2008*) and the second $\beta$-hairpin of the B1 domain of protein G (*Huyghues-Despointes et al., 2006*).

Our strategy for bioengineering new second generation SP-C mimics with high $\alpha$-helicity and surfactant properties primarily involves the introduction of $Glu^{-}$–$Lys^{+}$ pairings (i.e., salt-bridges or ion-locks) into SP-C sequences lacking palmitoyl groups. With a charged ion-pair in the midsection of the parent SP-Css (Fig. 1), SP-Css ion-lock 1 in either lipids, aqueous HFIP or aqueous buffer remarkably shifted its principal conformation to $\alpha$-helix from amyloid-like $\beta$-sheet (Fig. 2; Tables 1 and 2). In particular, FTIR spectral deconvolutions for SP-Css ion-lock 1 in surfactant lipids showed enhanced $\alpha$-helix similar to that determined for SP-C33 UCLA, our positive control SP-C peptide (i.e., 63.1 vs. 66.9%, respectively; Table 1). Oriented FTIR and H/D exchange experiments for SP-Css ion-lock 1 indicated a slightly tilted transmembrane $\alpha$-helix in surfactant lipid bilayers (Figs. 5 and 6; Table 1), consistent with MPEx hydropathy predictions for this SP-C mimic with a neutralized charged ion-pair (Fig. 4). A slightly tilted, transmembrane $\alpha$-helix was also identified for SP-C33 UCLA comparable to that of SP-Css ion-lock 1 (Figs. 5 and 6; Table 1). In this context, note that SP-Css ion-lock 1 showed high surfactant activities equal to SP-C33 UCLA, as measured with *in vitro* adsorption surface pressure (Fig. 7) and minimum surface tension (Fig. 8), and also with *in vivo* oxygenation and dynamic compliance experiments of surfactant-treated, ventilated rabbits with ARDS (Fig. 9). One explanation for the enhanced activities of SP-Css ion-lock 1 is that a close-range salt-bridge (*Kumar & Nussinov, 1999*; *Kumar & Nussinov, 2002*) forms between the $Glu^{-}$-20 and $Lys^{+}$-24 residues, which promotes accurate mimicking of the high $\alpha$-helicity and the membrane topography of SP-C33 UCLA (Fig. 2; Tables 1 and 2), SP-C33 (*Johansson et al., 2003*; *Almlen et al., 2011*) or native SP-C (*Pastrana, Maulone & Mendelsohn, 1991*; *Vandenbussche et al., 1992*). Analogous close-range salt-bridges occurring for the $Glu^{-}$-20 and $Lys^{+}$-24 pair might also account for the high $\alpha$-helix seen for SP-Css ion-lock 1 in either aqueous buffer or aqueous HFIP buffer (Table 2).

Experiments next evaluated correlations between $\alpha$-helicity, lipid bilayer topography and surfactant properties for the SP-Cff ion-lock 1 mimic. Although the insertion of the $Glu^{-}$–$Lys^{+}$ pairing into SP-Cff primarily converted the SP-Cff ion-lock 1 in lipids from $\beta$-sheet to $\alpha$-helix (Fig. 2A), FTIR deconvolutions of SP-Cff ion-lock 1 indicated lower $\alpha$-helix (i.e., 44.6%) than the corresponding $\alpha$-helix for SP-Css ion-lock 1 (63.1%) and SP-C33 UCLA (67.2%) (Table 1). Oriented FTIR and H/D exchange studies of SP-Cff ion-lock 1 showed that the $\alpha$-helix incorporates poorly into the lipid bilayer (Fig. 6), and also is more tilted (34.2°) to the membrane normal than those of either SP-Css ion-lock 1 (22.6°) or SP-C33 UCLA (22.4°) (Fig. 5; Table 1). Because of its shortened C-terminal

helix and much greater tilt, the $\alpha$-helix of SP-Cff ion-lock 1 may thus be restricted to a single monolayer. This abnormal membrane topography may be partially due to SP-Cff ion-lock 1 forming a disruptive, long-range salt-bridge (*Kumar & Nussinov, 2002*) between the $Glu^-$-20 and $Lys^+$-24 residues, which would be expected to reduce both helix stability (Table 1) and lipid affinity (Fig. 4). The exposure of the charged side-chain groups occurring in long-range salt-bridges (*Kumar & Nussinov, 2002*) may promote electrostatic interactions between the $Glu^-$-20 and $Lys^+$-24 residues of SP-Cff ion-lock 1 and other membrane components, including $Lys^+$-24 binding to anionic phosphates in the polar headgroup of POPG. Such detergent-like properties may be responsible for the low *in vitro* and *in vivo* surfactant activities obtained for formulations containing SP-Cff ion-lock 1, when compared with preparations containing SP-C33 UCLA, SP-Css ion-lock 1 or the parent SP-Cff (Figs. 7–9). Interestingly, SP-Cff ion-lock 1 in aqueous buffer additionally showed much lower $\alpha$-helix and higher loop-turns than those corresponding levels in SP-Css ion-lock 1 (Table 2), suggesting that replacement of the vicinal serines with phenylalanines may have also produced a long-range salt-bridge that was destabilizing in an aqueous environment.

In contrast to the surfactant properties of SP-Cff ion-lock 1, preparations containing SP-C33 ion-lock 1 showed elevated surface activity using captive bubble surfactometry (Fig. 8), and also high arterial oxygenation and dynamic compliance in surfactant-treated, ventilated rabbits with ARDS (Fig. 9). The enhanced *in vitro* and *in vivo* surfactant activities of SP-C33 ion-lock 1 match those of SP-Css ion-lock 1 and SP-C33 UCLA (Figs. 8 and 9), and are likely due to all three SP-C mimics exhibiting comparable $\alpha$-helicity in lipids (Fig. 2; Table 1) and incorporating as tilted, transmembrane $\alpha$-helices (Figs. 5 and 6; Table 1). For deep insertion into lipid bilayers as a helical transmembrane $\alpha$-helix, MPEx calculations in Fig. 4 additionally suggested that SP-C33 ion-lock 1 will form a close-range salt-bridge in the membrane interior. Interestingly, this electrostatically neutral ion-pair is unable to further increase the $\alpha$-helix content and surfactant activity of SP-C33 ion-lock 1 over those observed with the control SP-C33 UCLA, probably because these properties are already optimized in the host SP-C33 UCLA peptide. Nevertheless, the presence of a close-range salt-bridge in the mid-section of SP-C33 ion-lock 1 may improve both the thermostability (*Kumar, Tsai & Nussinov, 2000*; *Kumar & Nussinov, 2002*) and storage properties of this SP-C mimic over those of its parent SP-C33 UCLA.

Recent Molecular Dynamics (MD) simulations were also performed to assess further whether the enhanced surfactant activities for SP-Css ion-lock 1 were due to a close-range salt-bridge between $Glu^-$-20 and $Lys^+$-24 that stabilizes the $\alpha$-helix (AJ Waring, FJ Walther, KJ Longmuir, LM Gordon, unpublished data, 2014). MD runs using GRO-MACS (http://www.gromacs.org) should provide valuable information on the detailed 3D-structures of SP-C mimics in the bilayer (*Walther et al., 2010*; *Schwan et al., 2011*). For simulations of SP-Css ion-lock 1 in a lipid bilayer-water box, the "0 ns" model was obtained by first templating its primary sequence (Fig. 1) onto the 2D-NMR structure of porcine SP-C (*Johansson et al., 1994*) (PDB: 1SPF). The homology modelled SP-Css ion-lock 1 was centered in a bilayer box with its helical axis normal to the membrane surface, then

surrounded by 29 POPG lipids interacting with the transmembrane peptide as boundary lipid, and finally 79 DPPC lipids and 18 POPC lipids added to fill out the bilayer. In the GROMACS Version 4.5.5 environment, MD simulations (100 ns) for the SP-Css ion-lock 1-lipid ensemble in a water box using the ffG53a6 force-field rapidly reached equilibrium ($\leq$ ~60–70 ns). The equilibrated "100 ns" model shows that the hydrophobic C-terminus of SP-Css ion-lock 1 inserts as a transmembrane $\alpha$-helix (residues ~7–19 and ~22–32) at a slight tilt (i.e., $\theta$) to the normal of the bilayer surface, and also the polar N-terminus (residues ~1–6) interacts as an extended random-coil with polar lipid headgroups. In particular, the helical axis for the "100 ns" SP-Css ion-lock 1 is tilted at an angle (i.e., $\theta \sim 25°$) from the normal to the membrane surface that is in agreement with the maximum tilt angle ($\Theta$) from oriented FTIR spectra (i.e., ~23°) (see Table 1). The similarity in the membrane insertion angles for helical SP-Css ion-lock 1 obtained from MD simulation and polarized FTIR argues that the peptide-lipid bilayers are deposited as well-ordered multilayers on the germanium crystal with minimal unoriented (isotropic) peptide. Substantial amounts of isotropic peptide would be expected to reduce the dichroic ratio $R$ and increase $\Theta$ over the actual tilt angle $\theta$, which is not observed in our MD simulation and polarized FTIR experiments (Fig. 5; Table 1) (*Beevers & Kukol, 2006*). The MD simulations and "100 ns" models for SP-Css ion-lock 1, SP-C33 ion-lock 1 and SP-C33 are all similar, with the three SP-C mimics indicating comparable MD tilt angles (~20–25°) for their long (~22-residues) transmembrane $\alpha$-helices that match their polarized FTIR tilt angles ($\Theta = \sim 23°$) (see Table 1). Importantly, close-range salt-bridges were identified from the "100 ns" models for SP-Css ion-lock 1 and SP-C33 ion-lock 1 in the surfactant lipid bilayer, with gaps of 3.9 and 3.3 Å between the respective $Lys^+$-24 "N" and $Glu^-$-20 "O" pairs (*Kumar & Nussinov, 1999*; *Kumar & Nussinov, 2002*) (see Introduction). The salt-bridges for the SP-C mimics each lie at the center of the surfactant lipid bilayer, which is consistent with MPEx calculations indicating that SP-C peptides bearing ion-locks may readily embed into the lipid bilayer as $\alpha$-helices (Fig. 4). The relatively low H/D exchange for the $\alpha$-helical residues of either SP-C33 UCLA or SP-Css ion-lock 1 over ~6 h (Fig. 6) is also probably due to the deep penetration of these SP-C mimics into lipid bilayers as transmembrane $\alpha$-helices. Consequently, the high *in vitro* and *in vivo* surfactant activities reported here for SP-Css ion-lock 1, SP-C33 UCLA and SP-C33 ion-lock 1 (Figs. 7–9) are probably due to each of these SP-C mimics reproducing key structural features of native SP-C bound to lipid bilayers (see Introduction).

Analogous MD simulations were also conducted to determine whether the reduced surfactant activities of SP-Cff ion-lock 1 reported here were due to a long-range salt-bridge forming between $Glu^-$-20 and $Lys^+$-24 that destabilizes the $\alpha$-helix (AJ Waring, FJ Walther, KJ Longmuir & LM Gordon, 2014, unpublished data). Although the $Glu^-$–$Lys^+$ pair converts SP-Cff to principally $\alpha$-helical SP-Cff ion-lock 1 (Fig. 2A; Tables 1 and 2), FTIR deconvolutions of SP-Cff ion-lock 1 in lipids indicated lower $\alpha$-helix (i.e., 44.6%) than the $\alpha$-helix for SP-Css ion-lock 1 (63.1%) or SP-C33 UCLA (67.2%). Oriented FTIR and H/D exchange studies on SP-Cff ion-lock 1 further showed that its short $\alpha$-helix (~15 residues) poorly incorporated in the bilayer (Fig. 6), and also had a maximum tilt angle

($\Theta = 34.2°$) greater than those of SP-Css ion-lock 1 or SP-C33 UCLA (Fig. 5; Table 1). MD simulations confirmed that the "100 ns" SP-Cff ion-lock 1 restricted its shortened C-terminal $\alpha$-helix to one leaflet with a tilt ($\theta \sim 36°$) that matched the maximum tilt angle from polarized FTIR experiments. This abnormal membrane topography is partially due to SP-Cff ion-lock 1 forming a disruptive, long-range salt-bridge (8.9 Å) between the Glu$^-$-20 and Lys$^+$-24 residues (*Kumar & Nussinov, 2002*), which may reduce helix stability (Fig. 6) and lower lipid affinity (Fig. 4). The exposure of the charged side-chain groups occurring in long-range salt-bridges may promote electrostatic interactions between either Glu$^-$-20 or Lys$^+$-24 with other membrane components (*Kumar & Nussinov, 2002*), including Lys$^+$-24 binding to anionic phosphates in the polar headgroup of POPG. The trigger for the above "detergent-like" actions with SP-Cff ion-lock 1 is the replacement of the vicinal serines with phenylalanines. These substitutions significantly increase the hydrophobicity of the N-terminal domain of SP-Cff ion-lock 1 over that of SP-Css ion-lock 1, thereby permitting deeper insertion of the N-terminus into the fatty-acyl regions of the bilayer and 'snorkeling' of the C-terminal helix into the opposing monolayer. Such detergent-like properties may be responsible for the low *in vitro* and *in vivo* surfactant activities obtained for formulations containing SP-Cff ion-lock 1, when compared with preparations containing SP-C33 UCLA, SP-Css ion-lock 1 or SP-Cff (Figs. 7–9).

## CONCLUSIONS

Results from experiments on our suite of SP-C peptides confirm that a selective incorporation of salt-bridges produces SP-C mimics that insert into lipid bilayers as slightly tilted, transmembrane $\alpha$-helices with high surfactant activity. Our lead peptide is SP-Css ion-lock 1, an SP-C mimic that shows enhanced *in vitro* and *in vivo* surfactant activities over its parent SP-Css by readily inserting into lipid bilayers as an alpha-helix that is slightly tilted and spans both monolayers. Importantly, SP-Css ion-lock 1 simulates well both the structural and functional properties of SP-C33 UCLA, a member of the SP-C33 class of SP-C mimics. Our working hypothesis is that the SP-Css ion-lock 1 in lipids improves its surfactant functions by stabilizing $\alpha$-helix through a close-range salt-bridge between the Glu$^-$-20 and Lys$^+$-24 residues. As a further test of this proposition, more detailed 3D information on the structure and membrane topography of SP-C ion-locks will be obtained using $^{13}$C-FTIR spectroscopy (*Gordon et al., 2000*; *Waring et al., 2005*), 2D-NMR spectrometry (*Sarker et al., 2007*) and/or all-atom MD simulation techniques (*Walther et al., 2010*; *Almlen et al., 2010*). Additionally, polarized FTIR spectra of SP-C mimics in surfactant lipids should indicate the fatty-acyl chain orientation with respect to the peptide helical axis, and likewise show if there are any protein-induced perturbations in lipid conformations (*Clercx et al., 1995*). SP-C ion-lock peptides will also be modified to determine whether their surfactant properties may be further optimized. Because the palmitoyls of native SP-C have been proposed to increase surfactant activity by physically coupling stacked lipids to the monolayer (*Leonenko et al., 2007*), future atomic force microscopy (AFM) (*Ding et al., 2001*; *Frey et al., 2010*) and functional studies may assess if dipalmitoylated SP-Css ion-lock 1 similarly shows enhanced properties through an

analogous mechanism. Surface plasmon resonance (SPR) studies may also indicate if salt-bridges affect either the self-association of SP-C ion-locks or the binding of SP-C ion-locks to SP-B mimics or lipid ensembles (*Walther et al., 2010*).

Lastly, *in vivo* experiments will determine whether synthetic preparations containing SP-B analogs and SP-C ion-locks may be superior to single-peptide surfactants in the treatment of NRDS and ALI/ARDS. In earlier collaborative studies (*Almlen et al., 2010*), we investigated the *in vivo* activities of Mini-B (i.e., MB, a 34-residue 'short-cut' version of SP-B with an engineered turn linking the N- and C-terminal helical domains) and/or SP-C33 with synthetic lipids in preterm newborn rabbits. Treatment with either Mini-B or SP-C33 increased tidal and lung gas volumes, while combination therapy demonstrated an additive effect in this validated animal model for NRDS (*Almlen et al., 2010*). More recently, animal models of lung-lavaged ARDS and chemically induced ALI were treated with Super Mini-B (i.e., SMB, a 41-residue truncated version of SP-B, with the seven-residue N-terminus of SP-B added to the N-terminus of Mini-B) and SP-C33 UCLA. Synthetic formulations using either SMB or SP-C33 UCLA improved blood oxygenation and dynamic compliance in these ARDS/ALI models, while combination therapy indicated an additive effect (*Walther et al., 2014*). Similar *in vivo* experiments in which the SP-C33 class is replaced with SP-Css ion-lock 1 may show if SP-C mimics with salt-bridges may also be therapeutically beneficial.

**List of abbreviations**

| | |
|---|---|
| **ALI** | acute lung injury |
| **ARDS** | acute respiratory distress syndrome |
| **Asp** | aspartic acid |
| **Arg** | arginine |
| **ATR** | attenuated total reflectance |
| **CD** | circular dichroism |
| **DPPC** | dipalmitoyl phosphatidylcholine |
| **H/D** | hydrogen/deuterium |
| **HFIP** | hexafluoroisopropanol |
| **HPLC** | high performance liquid chromatography |
| **FTIR** | Fourier transform infrared spectroscopy |
| **Glu** | glutamate |
| **Lys** | lysine |
| **MPEx** | membrane protein explorer |
| **NRDS** | neonatal respiratory distress syndrome |
| **2D-NMR** | two-dimensional nuclear magnetic resonance spectroscopy |
| **PaO$_2$** | partial pressure of O$_2$ in arterial blood |
| **PaCO$_2$** | partial pressure of CO$_2$ in arterial blood |
| **PASTA** | prediction of amyloid structure aggregation |
| **POPC** | palmitoyl-oleoyl phosphatidylcholine |
| **POPG** | palmitoyl-oleoyl phosphatidylglycerol |

| | |
|---|---|
| **SP-B** | surfactant protein B |
| **SP-C** | surfactant protein C |
| **WW** | Wimley-White |

### Funding

The authors received financial support from the National Institutes of Health through grants R01HL092158 (FJW), R01ES015330 (FJW), and R01HL094641 (RHN). The NIH had no role in the design and conduct of the study, in the collection, analysis, and interpretation of the data, and in the preparation, review, or approval of the manuscript.

### Grant Disclosures

The following grant information was disclosed by the authors:
National Institutes of Health grants: R01HL092158, R01ES015330, R01HL094641.

### Competing Interests

FJW, AJW, LMG, ZW, and RHN have applied for a U.S. patent on "Novel SP-B & SP-C peptides, synthetic lung surfactant, and use thereof" relating to the ion-lock SP-C peptides described in this manuscript.

### Author Contributions

- Frans J. Walther conceived and designed the experiments, performed the experiments, analyzed the data, contributed reagents/materials/analysis tools, wrote the paper, prepared figures and/or tables, reviewed drafts of the paper.
- Alan J. Waring and Larry M. Gordon conceived and designed the experiments, performed the experiments, analyzed the data, wrote the paper, prepared figures and/or tables, reviewed drafts of the paper.
- José M. Hernández-Juviel, Piotr Ruchala and Zhengdong Wang performed the experiments, reviewed drafts of the paper.
- Robert H. Notter conceived and designed the experiments, analyzed the data, contributed reagents/materials/analysis tools, reviewed drafts of the paper.

### Animal Ethics

The following information was supplied relating to ethical approvals (i.e., approving body and any reference numbers):

The animal study was reviewed and approved by the Institutional Animal Care and Use Committee of the Los Angeles Biomedical Research Institute (protocol # 12958).

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
