# Peer review of "Surfactant protein C peptides with salt-bridges (“ion-locks”) promote high surfactant activities by mimicking the α-helix and membrane topography of the native protein"

_PeerJ, doi:10.7717/peerj.485_

## Round 0.1 · original submission · Minor Revisions

· Academic Editor

Minor Revisions

Please note comments made by Reviewer 1 regarding Fig 2 and 5 as well as Fig 6 regarding clarification of the orientation of the helix axis.

Although Reviewer 2 comments on some repetition in the discussion, I am in agreement that this is acceptable but should be noted.

·

Basic reporting

Any surfactant is a mixture of lipid and proteins that is crucial for normal breathing. Extensive effort has been made to replace bovine and porcine extracts with synthetic preparations made of proteins (SP-C and SP-B) and synthetic lipids. In a first step and to simplify peptide synthesis and purification, the covalently attached palmitoyl groups were omitted and replaced by serine and phenylalanine. This replacement however diminished strongly the in vitro surfactant activity probably as a consequence of a not correctly folded SP-C protein. The present work describes a new generation of surfactant but with enhanced in vitro and in vivo activities which is demonstrated to be associated to an enhanced helical content and modified membrane topology.
The article describes convincingly the main achievements and refers very appropriately and extensively to relevant literature. The introduction presents a chronological description of the progresses made in the development of more efficient surfactant preparations. It focuses nicely and specifically on the aspects discussed in the manuscript and relevant to the hypotheses being made.

Experimental design

Experiments are conducted very expertly and the methods used are exquisitely dedicated to the questions raised. There are a few minor comments that the authors may want to consider:

-Figure 2.There is no obvious reason to limit the wave number range to the protein domain and not to show the lipid domain (>1700 cm-1).This is also true for Figure 5.

-Figure 5- From a biochemical point of view, it would be more convincing to evaluate the helix axis orientation by referring to lipid orientation rather than to a normal to the germanium crystal. It is therefore important to confirm the orientation of the lipid in the SP-C-lipid complex.

-SP-C orientation raises another question. IR polarized spectra give access to a dipole orientation. This means that the helix axis orientation is strongly depending on the angle existing between the C=O bond orientation and the helix axis. This aspect deserves to be clarified.

-Any estimation of the dichroic ratio should consider the isotropic contribution.This possibility is not discussed in the manuscript.

-Figure 6- Percentages of SP-Cff ion-lock 1 unexchanged hydrogens suggest that the hydrophobic domain of the peptide which is inserted into the lipid bilayer is largely accessible to the water environment. Does it mean that a slight change of orientation of the lipid (37 degrees) makes it almost fully accessible to the water environment.

Validity of the findings

The strategy for building a new generation of surfactant is convincing and based on robust and statistically sound data.The correlation between helicity, lipid bilayer topography and in vitro and in vivo activity is impressive and opens the way to the rational design of even more active surfactant preparations.

Reviewer 2 ·

Basic reporting

No Comments

Experimental design

No Comments

Validity of the findings

No Comment

Additional comments

This article applies a wide range of tools to interrogate novel SP-C peptides secondary structure and orientation in synthetic surfactant lipid mixtures. The results and conclusions regarding the advantages of the ion-lock design are substantiated. I did think that that overall the discussion presented was too repetitive because it took place in the intro, results, and discussion sections. However, I don't think it would be worth the author's time at this point to condense the text.

---

## Round 0.2 · accepted · Accept

· Academic Editor

Accept

Look forward to the MD paper.

·

Basic reporting

No comments

Experimental design

No comments

Validity of the findings

No comments

Additional comments

The revised version addresses convincingly most criticisms.The references to unpublished results strengthen the impact of the manuscript